# How Mask Matters: Towards Theoretical Understandings of Masked Autoencoders

**Qi Zhang**[1*]       **Yifei Wang**[2*]       **Yisen Wang**[1,3†]

[1] Key Lab. of Machine Perception (MoE),
School of Intelligence Science and Technology, Peking University
[2] School of Mathematical Sciences, Peking University
[3] Institute for Artificial Intelligence, Peking University

## Abstract

Masked Autoencoders (MAE) based on a reconstruction task have risen to be a promising paradigm for self-supervised learning (SSL) and achieve state-of-the-art performance across different benchmark datasets. However, despite its impressive empirical success, there is still limited theoretical understanding of it. In this paper, we propose a theoretical understanding of how masking matters for MAE to learn meaningful features. We establish a close connection between MAE and contrastive learning, which shows that MAE implicit aligns the mask-induced positive pairs. Built upon this connection, we develop the first downstream guarantees for MAE methods, and analyze the effect of mask ratio. Besides, as a result of the implicit alignment, we also point out the dimensional collapse issue of MAE, and propose a Uniformity-enhanced MAE (U-MAE) loss that can effectively address this issue and bring significant improvements on real-world datasets, including CIFAR-10, ImageNet-100, and ImageNet-1K. Code is available at https://github.com/zhangq327/U-MAE.

## 1 Introduction

Recently, self-supervised learning (SSL) has been proposed as a promising paradigm for learning data representations without access to labels. Beside the popular contrastive learning methods [4, 14, 12, 27], recently there has been a renaissance of reconstruction-based autoencoders for SSL, for example, MAE [15], BEiT [1], iBOT [32], and SimMIM [29], which demonstrate state-of-the-art performance on various downstream tasks. Taking MAE as an example, it learns to reconstruct the masked patches from the unmasked context with an encoder-decoder architecture, from which we can see that there are two main components in MAE: masking and autoencoders. While for autoencoders, its canonical one, even equipped with expressive neural networks, is still less competitive than modern SSL methods [4], which indicates that autoencoders may not be the key factor for the success of MAE. Then for masking, MAE needs a very large mask ratio, *e.g.,* 75% patches of the input image are masked out. Such a large mask ratio will lead to the loss of most image contents that are hardly recoverable from the rest. Further considering that MAE deliberately chooses a weak decoder with a shadow architecture, we believe that reconstruction might not be the ultimate goal of MAE in the learning process. Then, some natural questions are raised here:

*What is the role of masking in MAE? How does it affect the downstream performance?*

To answer these questions, firstly, we build a close connection between MAE and contrastive learning. In particular, we find that masking could produce *implicit* positive pairs while MAE's reconstruction

---

[*]Equal Contribution.

[†]Corresponding author: Yisen Wang (yisen.wang@pku.edu.cn).

loss is lower bounded by an *alignment* loss on these positive pairs. Further for feature uniformity, we notice that although MAE will not fully collapse, it still suffers from *dimensional collapse* where features lie in a low-dimensional space and become highly alike. Inspired by the uniformity loss in contrastive learning, we propose a Uniformity-enhanced MAE (**U-MAE**) to promote the feature diversity. Empirically, the proposed U-MAE attains significant improvements over MAE on the linear probing task across different benchmark datasets (CIFAR-10, ImageNet-100, and ImageNet-1K) and different ViT backbones, and it effectively eliminates the feature collapse issue of MAE. Secondly, the connection between MAE and contrastive learning enables us to establish the *first theoretical guarantee* for downstream classification among MAE methods. Our guarantees suggest that a large mask ratio is indeed necessary for bridging semantically similar samples together.

We summarize our contributions as follows:

- We propose a new theoretical understanding of MAE by establishing a formal connection between MAE and contrastive learning. In particular, we show that a small reconstruction loss implies better alignment of mask-induced positive pairs.
- Built upon this connection, we establish the first theoretical guarantee on the downstream performance among MAE methods, which helps to understand the role of high mask ratio.
- We point out the dimensional collapse issue of MAE, and propose U-MAE that enhances feature diversity with a uniformity loss. Empirically, U-MAE improves the linear probing accuracy of MAE by a large margin across different real-world datasets and backbones.

## 2  Related work

**Self-Supervised Learning.** Canonical deep learning relies on labeled data (*e.g.,* ImageNet) to train deep models. Instead, self-supervised learning (SSL) aims to learn meaningful representations from fully unlabeled data. A particular example is contrastive learning (CL) that learns to align augmented samples in the feature space, which achieves remarkable success and largely closes the performance gap between supervised and self-supervised learning [4, 14, 12, 27]. Recently, Masked Image Modeling (MIM), stemming from the Masked Language Modeling (MLM) paradigm widely adopted in NLP (*e.g.,* BERT [9]), also shows promising results in visual representation learning, to name a few, BEiT [1], iBOT [32], SimMIM [29], and MAE [15]. Among these MIM methods, the former three (similar to BERT) have close connections to contrastive learning (their objective can be directly formulated as contrastive loss [20]), while MAE has several key differences to BERT-like methods. As the name suggests, MAE is more like an autoencoder rather than a token predictor: it adopts a *pixel-level* reconstruction loss, omits the *masked tokens* in the encoder input, and utilizes a fully *non-linear* encoder-decoder architecture. In this work, we mainly provide a theoretical understanding of the working mechanism of MAE, in particular, how it learns generalizable features. Meanwhile, our analysis can also be applied to other MIM methods by different specifications of the encoder-decoder and their inputs.

**Understanding MAE.** Despite the impressive success of MAE, the theoretical understanding of it is largely underexplored. Among several concurrent attempts, Cao *et al.* [2] mostly focus on the attention mechanism of MAE through an integral kernel perspective. A similar theoretical analysis [23] also shows that autoencoders can preserve useful semantics in the pretraining data. However, we notice that these MAE analyses mostly focus on the autoencoder architecture, while they largely ignore the role of masking. As shown by He *et al.* [15], the patchwise masking strategy is a key component that distinguishes MAE from standard autoencoders, and different mask ratios $\rho$ have a large impact on the downstream performance of pretrained features of MAE. Therefore, our work aims to explain how the masking-based reconstruction task learns meaningful representations.

**Downstream Generalization of SSL.** Motivated by the empirical success of SSL, many researchers try to theoretically understand how SSL works. Arora *et al.* [25] establish downstream guarantees of contrastive learning representations. Wang *et al.* [28] revise their bounds by resolving the class collision issues, and develop a new understanding of contrastive learning from the perspective of augmentation overlap. Haochen *et al.* [13] propose an augmentation graph framework and characterize the downstream performance of the eigen-decomposition solution. While these prior works focus on the contrastive learning method, there are little theoretical understanding on the downstream performance on MAE. In this work, we establish the first theoretical guarantee on the downstream performance among MAE methods and discuss the effect of mask ratios theoretically and empirically.

# 3 Masked Autoencoders Perform Implicit Contrastive Learning

In this section, we establish a formal connection between MAE and contrastive learning by showing that MAE implicitly aligns positive input pairs induced from the masking mechanism. Specifically, in Section 3.2, we show how a small MAE loss implies a small alignment loss. Motivated by this connection, in Section 3.3, we further study its feature collapse issue and point out that MAE suffers from dimensional feature collapse. To address this issue, in Section 3.4, we propose a Uniformity-enhanced (U-MAE) loss to further promote feature diversity.

## 3.1 Mathematical Formulation for MAE

We begin by introducing a mathematical formulation of MAE [15]. Given a natural image $\bar{x}$ from an unlabeled dataset $\mathcal{D}_u$, we first reshape it into $n$ patches, denoted as $\bar{x} \in \mathbb{R}^{n \times s}$ where $s$ denotes the patch size (*e.g.,* $16 \times 16$ in ViT [10]). Then, we draw a random binary mask $m \in \{0, 1\}^n$ (drawing 0 with probability $\rho$, *i.e.,* the mask ratio), and obtain two complementary masked views of $\bar{x}$:

$$x_1 = \bar{x}[m] \in \mathbb{R}^{n_1 \times s}, \quad x_2 = \bar{x}[1 - m] \in \mathbb{R}^{n_2 \times s}, \tag{1}$$

where $n_1 = n(1 - \rho), n_2 = n\rho$ are integers satisfying $n = n_1 + n_2$. We denote this random masking process as drawing $x_1, x_2$ from the joint distribution $\mathcal{M}(x_1, x_2 | \bar{x})$, whose marginal distributions are $\mathcal{M}_1(x_1 | \bar{x})$ and $\mathcal{M}_2(x_2 | \bar{x})$, respectively. The MAE model $h = g \circ f$ is an encoder-decoder architecture, where an encoder $f$ maps inputs $x_1$ to a latent feature $z_1 = f(x_1)$, and a decoder $g$ maps the latent feature $z_1$ back to the pixel space to reconstruct the complementary target view $x_2$. Specifically, MAE adopts a simple mean square error (MSE) loss:

$$\mathcal{L}_{\text{MAE}}(h) = \mathbb{E}_{\bar{x}} \mathbb{E}_{x_1, x_2 | \bar{x}} \| g(f(x_1)) - x_2 \|^2, \tag{2}$$

where the decoder output $\hat{x}_2 = h(x_1) = g(f(x_1))$ is assumed to be $l_2$-normalized following the original paper of MAE saying that normalized target $x_2$ yields better performance [15]. Although our analysis is based on MAE, it is quite general and can be naturally transferred to other Masked Image Modeling (MIM) frameworks, such as BEiT [1], iBOT [32], and SimMIM [29]. This is because their differences mainly lie in the implementation details, which is discussed in detail in Appendix C.

**The Bipartite Mask Graph $\mathcal{G}_M$ of MAE.** We notice that MAE essentially learns to pair the two complementary views $x_1, x_2$ via the reconstruction task, which can be modeled by a *mask graph* $\mathcal{G}_M$. Denote the set of all unmasked views as $\mathcal{X}_1 = \{x_1\}$ and the set of all masked views as $\mathcal{X}_2 = \{x_2\}$, where the two sets are assumed to be finite[3] (can be exponentially large), *i.e.,* $|\mathcal{X}_1| = N_1, |\mathcal{X}_2| = N_2$. The mask graph $\mathcal{G}_M$ over the joint set $\mathcal{X} = \mathcal{X}_1 \cup \mathcal{X}_2$ is defined here:

- Node: each view $x \in \mathcal{X}$.
- Edge: the edge weight $w_{x_1, x_2}$ between any $x_1, x_2 \in \mathcal{X}$ is defined as their joint probability $\mathcal{M}(x_1, x_2) = \mathbb{E}_{\bar{x}} \mathcal{M}(x_1, x_2 | \bar{x})$. In other words, there is an edge between two views (*i.e.,* $w_{x_1, x_2} > 0$) if and only if they are complementary views generated by masking.

Considering the masking process in Eq. 1, there only exist edges *between* the two sets $\mathcal{X}_1, \mathcal{X}_2$ and there is no edge *within* each set itself. Thus, the mask graph $\mathcal{G}_M$ is a *bipartite* graph. Its adjacency matrix can be simply defined as $A_M \in \mathbb{R}^{N_2 \times N_1}$ where $(A_M)_{x_2, x_1} = w_{x_1, x_2}$ for $x_1 \in \mathcal{X}_1, x_2 \in \mathcal{X}_2$. As long as $\rho \neq 0.5$, $N_1 \neq N_2$ and $A_M$ is not necessarily a square matrix. We can define the normalized adjacency matrix as $\bar{A}_M = D_2^{-1/2} A D_1^{-1/2}$, where $D_1, D_2$ are the diagonal degree matrices with elements $d_{x_1} = \sum_{x_2} w_{x_1, x_2}$ and $d_{x_2} = \sum_{x_1} w_{x_1, x_2}$, respectively.

## 3.2 MAE Implicitly Aligns Positive Input Pairs

As the name suggests, MAE is basically composed of two key designs: mask and autoencoders. For simplicity, we begin by assuming that the encoder-decoder architecture of MAE is capable of accomplishing the *vanilla* autoencoder task, *i.e.,* reconstructing the original input.

**Assumption 3.1.** *For any non-degenerate decoder $g$, we assume the existence of a pseudo-inverse encoder $f_g \in \mathcal{F}$ such that the resulting pseudo autoencoder $h_g = g \circ f_g$ satisfies $\mathbb{E}_x \| h_g(x) - x \|^2 \leq \varepsilon$, where $x$ represents either unmasked data $x_1$ or masked data $x_2$.*

---

[3]This is used to avoid non-essential nuances related to functional analysis. Our discussion can also be extended to the infinite data regime following Haochen *et al.* [13].

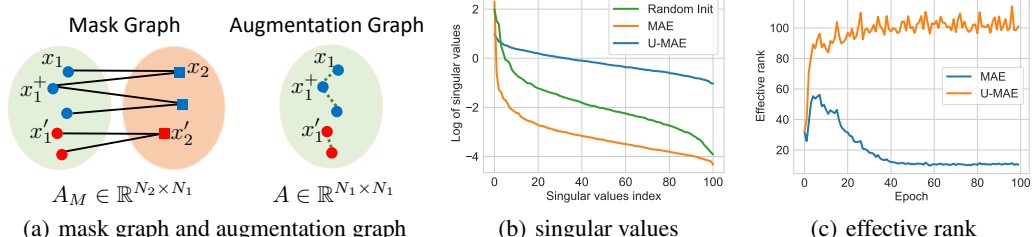

Figure 1: (a) An illustration of the mask graph and the corresponding augmentation graph of MAE. Different colors denote different belonging classes. (b) Comparison of the singular values of learned features with 1) random initialization, 2) MAE loss, 3) our U-MAE loss. (c) The changing process of effective rank [24] of the encoded features trained with different objectives (MAE and U-MAE).

This assumption is likely to hold in practice, since deep networks already demonstrate excellent performance on the autoencoding task [19, 17]. Besides, some works show that the Transformer network adopted in MAE has universal approximation ability [30]. However, we note that vanilla autoencoder task cannot learn as meaningful representations as MAE. For example, without applying any masks, the linear probing accuracy of MAE drops dramatically from 61.2% to 17.4% on ImageNet-100. This indicates that the autoencoding ability (studied in [2, 23]) is *not* enough to explain the efficacy of MAE, which motivates us to further explore the masking mechanism in MAE.

**MAE Performs Asymmetric Input-output Alignment on the Mask Graph.** First, we show that the MAE loss can be lower bounded by an *asymmetric* alignment loss between the two complementary views $x_1, x_2$ using two-branch autoencoders $h$ (real) and $h_g$ (pseudo), respectively.

**Theorem 3.2.** *Under Assumption 3.1, the MAE loss can be lower bounded by*

$$\mathcal{L}_{MAE}(h) \geq \mathcal{L}_{asym}(h) - \varepsilon + const, \tag{3}$$

$$and\ \mathcal{L}_{asym}(h) = -\mathbb{E}_{x_1,x_2} h(x_1)^\top h_g(x_2) = -\operatorname{tr}(H^\top \bar{A}_M H_g), \tag{4}$$

*where $H$ denotes the output matrix of $h$ on $\mathcal{X}_1$ whose $x_1$-th row is $H_{x_1} = \sqrt{d_{x_1}} h(x_1)$, and $H_g$ denotes the output matrix of $h_g$ on $\mathcal{X}_2$ whose $x_2$-th row is $(H_g)_{x_2} = \sqrt{d_{x_2}} h_g(x_2)$.*

As a result, a small MAE loss implies a small alignment loss (as a lower bound), which indicates that MAE will implicitly align the masked and unmasked views with an encoder-decoder architecture. However, it is a little bit confusing why aligning the two complementary views helps learn meaningful features. For a closer look, we find that via masking, MAE actually produces *implicit connections* among different *input samples* in the form of *2-hop connectivity*. Consider a pair of 2-hop input neighbors $x_1, x_1^+ \in \mathcal{X}_1$ that share a common complementary target view $x_2 \in \mathcal{X}_2$ (more likely to happen under a larger mask ratio $\rho$), as illustrated in Figure 1(a). By enforcing $x_1, x_1^+$ to reconstruct the same output $x_2$, MAE will implicitly map their features together. In this way, the 2-hop input neighbors serve as *positive pairs* that will be implicitly aligned as in contrastive learning.

Motivated by this observation, we seek to establish a formal connection between MAE and contrastive learning below. In particular, we will show that like the data augmentations in contrastive learning, the masking mechanism of MAE also implicitly introduces affinities between *input samples* and create (implicit) *positive pairs*. For a formal exposure, we define an augmentation graph $\mathcal{G}_A$ for modeling the relationship between all input samples in $\mathcal{X}_1$, which is different from the mask graph $\mathcal{G}_M$ for modeling the input-output relationship between $\mathcal{X}_1$ and $\mathcal{X}_2$.

**The Augmentation Graph $\mathcal{G}_A$ of MAE.** We construct an *augmentation graph* $\mathcal{G}_A$ for modeling the *mask-induced affinity* between all unmasked views in $\mathcal{X}_1$ [4]. Specifically, for any two views $x_1, x_1^+ \in \mathcal{X}_1$, we define their edge weight in the adjacency matrix $A$ as the probability of having the same target view, *i.e.*, $\mathcal{A}(x_1, x_1') = \mathbb{E}_{x_2} \mathcal{M}(x_1|x_2)\mathcal{M}(x_1'|x_2)$ [5]. Note that the augmentation graph

---

[4]We can also construct an augmentation graph $\mathcal{G}_A'$ on the output space $\mathcal{X}_2$, where the $(x_2, x_2')$-th element of the adjacency matrix $A'$ is defined by $\mathcal{A}(x_2, x_2') = \mathbb{E}_{x_1} \mathcal{M}(x_2|x_1)\mathcal{M}(x_2'|x_1)$. As the results are quite similar, we mainly take the input space as an example in this work.

[5]$\mathcal{M}(x_1|x_2) = \mathcal{M}(x_1, x_2)/\mathcal{M}(x_2)$ can further be calculated by marginalizing $\mathcal{M}(x_1, x_2|\bar{x})$.

has the same degree matrix $D_1$ as the mask graph. Based on this formulation, we define a *symmetric alignment loss w.r.t.* the positive pairs $(x_1, x_1^+) \sim \mathcal{A}(x_1, x_1^+)$ drawn according to the augmentation graph:

$$\mathcal{L}_{\text{align}}(h) = -\mathbb{E}_{x_1, x_1^+} h(x_1)^\top h(x_1^+). \tag{5}$$

**MAE Performs Symmetric Input Alignment on the Augmentation Graph.** Built upon the augmentation graph constructed above, we theoretically verify the intuition on 2-hop connectivity by establishing the relationship between the asymmetric input-output alignment loss on the mask graph and the symmetric input alignment loss on the augmentation graph in the following theorem.

**Theorem 3.3.** *The asymmetric alignment loss on the mask graph (Eq. (4)) can be lower bounded by the symmetric alignment loss on the augmentation graph (Eq. (5)):*

$$\mathcal{L}_{asym}(h) \geq \frac{1}{2}\mathcal{L}_{align}(h) + const. \tag{6}$$

***Proof Sketch.*** *We provide a proof sketch of this inequality as it is the key to our analysis. We first symmetrize the asymmetric alignment loss $\mathcal{L}_{asym}(h)$ with an arithmetic inequality, and then establish its equivalence to the symmetric alignment loss $\mathcal{L}_{align}(h)$. The derivation highlights the intrinsic connection between the mask graph (adjacency matrix $A_M$) and the augmentation graph (adjacency matrix $A$).*

$$
\begin{aligned}
\mathcal{L}_{asym}(h) &= \mathbb{E}_{x_1, x_2} h(x_1)^\top h_g(x_2) \\
&= -\text{tr}(H^\top \bar{A}_M H_g) &&\textit{(reformulated to the mask graph (Theorem 3.2))} \\
&\geq -\frac{1}{2}(\|H^\top \bar{A}_M\|^2 + \|H_g\|^2) &&\textit{(because } \text{tr}(AB) \leq \frac{1}{2}(\|A\|^2 + \|B\|^2)) \\
&= -\frac{1}{2}\text{tr}(H^\top A_M A_M^\top H) - \frac{1}{2} &&\textit{(because } \|H_g\|^2 = \sum_{x_2} d_{x_2}\|h_g(x_2)\|^2 = 1) \\
&= -\frac{1}{2}\sum_{x_1, x_1'} A_{x_1, x_1'} h(x_1)^\top h(x_1') - \frac{1}{2} &&\textit{(transformed to the augmentation graph)} \\
&= \frac{1}{2}\mathcal{L}_{align}(h) - \frac{1}{2}. &&\textit{(following the definition in Eq. (5))}
\end{aligned}
$$

Combining Theorem 3.2 and Theorem 3.3, we arrive at the main theorem showing that MAE's reconstruction loss can be bounded by the symmetric alignment loss of the positive input pairs $(x_1, x_1^+)$ drawn according to the augmentation graph.

**Theorem 3.4.** *Under Assumption 3.1, MAE's reconstruction loss (Eq. (2)) can be lower bounded by the alignment loss between positive pairs $(x_1, x_1^+) \sim \mathcal{A}(x_1, x_1^+)$,*

$$\mathcal{L}_{MAE}(h) \geq \frac{1}{2}\mathcal{L}_{align}(h) - \varepsilon + const = -\frac{1}{2}\mathbb{E}_{x_1, x_1^+} h(x_1)^\top h(x_1^+) - \varepsilon + const. \tag{7}$$

In this way, we establish a close relationship between the two mainstream SSL paradigms (MAE and contrastive learning) by showing that a small MAE loss will imply a small alignment loss of positive input pairs as in contrastive learning. Leveraging this connection, we could establish guarantees for the downstream generalization of MAE (discussed in Section 4).

**Comparison to Contrastive Learning.** Comparing the alignment loss (Eq. (5)) to that of contrastive learning [22, 4, 13], we notice that the main difference lies in that contrastive learning aligns features in the *latent* space of the encoder $f$, while MAE aligns features in the *output* space with an encoder-decoder architecture $h = g \circ f$. Nevertheless, we also note that most variants of contrastive learning apply a nonlinear projection head $g$ upon the encoder $f$ before calculating the alignment loss [4, 5, 14, 12, 6], and MAE also adopts a shadow decoder $g$. Therefore, we may also regard the role of MAE's decoder as the projection head in contrastive learning. Theoretically, if we further assume the bi-Lipschitzness of the decoder, we can show that the MAE loss is further lower bounded by an alignment loss defined in the feature space.

**Corollary 3.5.** *Under Assumption 3.1 and the assumption that the decoder is $L$-bi-Lipschitz, i.e., $\forall (x_1, x_2), 1/L\|x_1 - x_2\|^2 \leq \|g(x_1) - g(x_2)\|^2 \leq L\|x_1 - x_2\|^2$. Then, the MAE reconstruction loss can be lower bounded by the alignment loss w.r.t. the encoder outputs:*

$$\mathcal{L}_{MAE}(h) \geq -1/(2L) \cdot \mathbb{E}_{x_1, x_1^+} f(x_1)^\top f(x_1^+) - \varepsilon + const. \tag{8}$$

### 3.3 The Feature Collapse Issue in MAE

We have revealed that the MAE loss is closely related to an alignment loss. However, it is well known that in contrastive learning, simply aligning the positive pairs will result in the full feature collapse, because the alignment loss can also be minimized when the encoder produces a constant feature for all inputs. There, various techniques are proposed to address the issue by incorporating an additional loss to encourage feature uniformity or feature decorrelation [4, 14, 31], or by asymmetric structural designs [12, 6, 3]. MAE does not implement any of these techniques, but its latent features still do not fully collapse. One would wonder how MAE attains this property. In the following theorem, we show that minimizing the MAE loss can provably get rid of the full feature collapse.

**Theorem 3.6.** *When the encoder fully collapses, i.e., $\forall x \in \mathcal{X}_1, f(x) = c$, the MAE loss has a large lower bound:*

$$\mathcal{L}_{MAE}(h) \geq \mathrm{Var}(x_2), \tag{9}$$

*where $\mathrm{Var}(x_2)$ denotes the variance of masked targets computed on the training dataset.*

**MAE Can Avoid Full Feature Collapse.** As the training data contain diverse images, the variance $\mathrm{Var}(x_2)$ will be relatively large. Therefore, unlike the alignment loss in contrastive learning, the MAE loss cannot be minimized (to a small value) by a collapsed encoder. The main reason is that alignment loss operates in a fully flexible latent space that allows a collapsed encoder to minimize the loss, while in MAE, the reconstruction loss adopts a parameter-invariant and sample-dependent target $x_2$. In this case, a fully collapsed encoder can no longer minimize the reconstruction loss *w.r.t.* $x_2$, which, in turn, means that MAE will not fully collapse.

**MAE Still Suffers from Dimensional Collapse.** Although MAE can avoid full feature collapse, it could still suffer from *dimensional feature collapse* where the learned features lie in a low dimensional subspace [16, 18], which also limits its representation power. We empirically examine this issue on ImageNet-100. Figure 1(b) shows that after learning, MAE's features indeed become more collapsed as there are fewer large singular values (indicating non-collapse dimensions). Quantitatively, Figure 1(c) shows that the features of MAE gradually collapse during the training process, as the effective rank [24] becomes smaller and smaller. This shows that MAE suffers from an increasing degree of dimensional feature collapse. Next, we introduce an explicit regularization to address this issue.

### 3.4 U-MAE: Enhancing Feature Diversity with an Explicit Uniformity Regularization

To further enhance the feature diversity of MAE, inspired by the uniformity loss in contrastive learning [22, 4, 13], we propose the following Uniformity-enhanced MAE (U-MAE) loss with an explicit regularization on feature uniformity through a coefficient $\lambda > 0$,

$$\mathcal{L}_{\text{U-MAE}}(h) = \mathcal{L}_{\text{MAE}}(h) + \lambda \cdot \mathcal{L}_{\text{unif}}(f), \tag{10}$$

$$\text{where } \mathcal{L}_{\text{unif}}(f) = \mathbb{E}_{x_1}\mathbb{E}_{x_1^-}(f(x_1)^\top f(x_1^-))^2, \tag{11}$$

and $x_1^-$ denotes an independently drawn unmasked view from $\mathcal{X}_1$. Intuitively, the spectral uniformity loss $\mathcal{L}_{\text{unif}}(f)$ [13] encourages a small feature similarity between random unmasked views, which could effectively promote the feature diversity of all samples. As a preview of the results, we can observe from Figures 1(b) & 1(c) that U-MAE effectively addresses the dimensional feature collapse issue and improves the effective feature dimensionality by a large margin.

Theoretically, combined with Corollary 3.5, we can show that the U-MAE loss is lower-bounded by the spectral contrastive loss with a specific choice of $\lambda$.

**Theorem 3.7.** *Denote the spectral contrastive loss (SCL) from Haochen et al. [13] as*

$$\mathcal{L}_{SCL}(f) = 2\mathcal{L}_{align}(f) + \mathcal{L}_{unif}(f) = -2\mathbb{E}_{x_1,x_1^+}f(x_1)^\top f(x_1^+) + \mathbb{E}_{x_1,x_1'}(f(x_1)^\top f(x_1^-))^2. \tag{12}$$

*Under Assumption 3.1 and the assumption that the decoder is L-bi-Lipschitz, when $\lambda = 1/(4L)$, the U-MAE loss can be lower bounded by the SCL loss:*

$$\mathcal{L}_{U\text{-}MAE}(h) \geq \frac{1}{4L} \cdot \mathcal{L}_{SCL}(f) - \varepsilon + const. \tag{13}$$

As a result, minimizing the U-MAE loss will implicitly minimize the spectral contrastive loss among input views. Leveraging this inherent connection, in the next section, we further explain the important role of masking on the downstream generalization of MAE (as well as U-MAE).

# 4 Downstream Generalization of Masked Autoencoders

In Section 3, we develop a close connection between MAE and contrastive learning. Built upon this connection, in Section 4.1, we establish theoretical guarantees on the downstream performance of U-MAE leveraging the mask-induced augmentation graph introduced in Section 3.2. Built upon these theoretical insights, in Section 4.2, we further investigate the effect of mask ratios and explain the optimal mask ratio of MAE.

## 4.1 Theoretical Guarantees on Downstream Classification

In this part, we characterize the downstream performance of U-MAE on the $c$-class linear classification task [25, 13], which is measured by the prediction accuracy of natural images $\bar{x}$ *w.r.t.* their labels $y(\bar{x})$ when applying a linear prediction head upon pretrained features. For simplicity, we adopt the mean classifier $p_f(x) = \arg\max W_f f(x)$, where $W_f \in \mathbb{R}^{c \times k}$ is the weight of the linear classification head, and for $y \in [c]$, the $y$-th row $W_y = \mathbb{E}_{x_1|y} f(x_1)^\top$ contains the mean representation of the class $y$. Arora *et al.* [25] empirically show that the mean classifier can obtain comparable performance to learnable linear heads. By default, we set the uniformity coefficient $\lambda = 1/(4L)$ as in Theorem 3.7.

**Theorem 4.1.** *Denote the mask-induced label error as $\alpha = \mathbb{E}_{\bar{x}, x_1} \mathbb{1}[y(x_1) \neq y(\bar{x})]$. Then, for $\forall\, h \in \mathcal{H}$ (the hypothesis class) with $h = g \circ f$, the downstream classification error of its encoder can be upper bounded by its U-MAE pretraining loss:*

$$\Pr(\bar{y} \neq p_f(\bar{x})) \leq c_1 L \cdot \mathcal{L}_{U\text{-}MAE}(h) + c_2\alpha + c_3 L\varepsilon + c_4, \tag{14}$$

*where $c_1, \ldots, c_4$ are constants and $c_3 > 1$.*

This theorem provides an upper bound on the downstream error of an encoder $f$ with its U-MAE pretraining loss, which is the *first theoretical guarantee* on the downstream performance of MAE methods. As an implication of this theorem, a small U-MAE loss would provably imply a small downstream classification error, which helps explain the good downstream generalization ability of MAE [15]. Besides, we establish a common lower bound on the U-MAE loss that holds for all $h \in \mathcal{H}$. As a large common lower bound of U-MAE loss indicates that the downstream error will always have a large upper bound, we should pursue a small common lower bound in the following theorem.

**Theorem 4.2.** *The U-MAE pretraining loss has the following common lower bound:*

$$\forall\, h \in \mathcal{H}, \quad \mathcal{L}_{U\text{-}MAE}(h) \geq \frac{1}{4L} \sum_{i=k+1}^{N_1} \lambda_i^2 - \varepsilon + const, \tag{15}$$

*where $\lambda_1 \geq \cdots \geq \lambda_{N_1}$ denote the eigenvalues of $A$.*

Combining Theorem 4.1 and Theorem 4.2, we can see that the downstream error of MAE can be minimized with a small vanilla autoencoding error $\varepsilon$, a small label error $\alpha$, and smaller magnitude of "residual eigenvalues", *i.e.*, $\{\lambda_{k+1}, \ldots, \lambda_{N_1}\}$. Here, residual eigenvalues represent the high frequency components of the augmentation graph $\mathcal{G}_A$ that cannot be fitted by $k$-dimensional features. The vanilla autoencoding error $\varepsilon$ depends on the capacity of the chosen model class $\mathcal{H}$, while the label error $\alpha$ and residual eigenvalues $\{\lambda_{k+1}, \ldots, \lambda_{N_1}\}$ purely depend on the mask-induced augmentation graph $\mathcal{G}_A$. Therefore, overall speaking, MAE with large capacity can have a smaller downstream error with a smaller autoencoding error $\varepsilon$. In the meantime, the masking ratio $\rho$ should be properly chosen to ensure a small label error $\alpha$ and small magnitude of residual eigenvalues $\{\lambda_{k+1}, \ldots, \lambda_{N_1}\}$. Indeed, He *et al.* [15] show that the mask ratio has a decisive influence on the downstream performance of MAE. Therefore, in the next part, we discuss how the choice of mask ratio would affect the downstream generalization of MAE via the label error $\alpha$ and residual eigenvalues $\{\lambda_{k+1}, \ldots, \lambda_{N_1}\}$.

## 4.2 The Effect of Mask Ratio on Downstream Generalization

A surprising fact in MAE is that the optimal mask ratio is pretty high, *e.g.*, $\rho = 0.75$, where most image contents are lost. Therefore, we are wondering why such a high mask ratio is necessary and how it helps (instead of damaging) the learning of data representations. In this part, we provide both theoretical and empirical insights on this problem based on our proposed theory.

**Theoretical Insights.** As discussed above, Theorems 4.1 & 4.2 suggest that a small label error $\alpha$ and small magnitude of residual eigenvalues $\{\lambda_{k+1}, \ldots, \lambda_{N_1}\}$ are important for good downstream

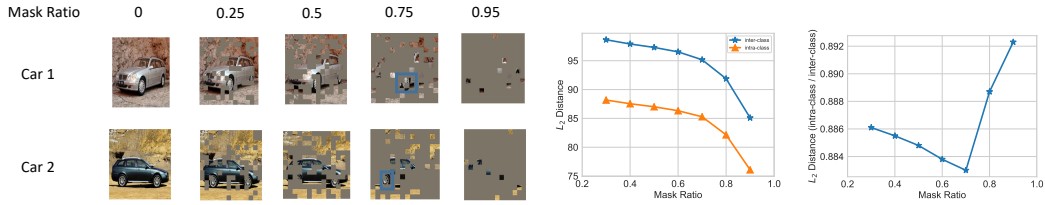

(a) Masked views with different mask ratio

(b) The distance between intra-class and inter-class samples

(c) The relative distance between intra-class samples and inter-class samples

Figure 2: (a) Appropriate mask ratio can generate similar views from different samples in the same class. (b) The influence of mask ratio on the average $l_2$ distance on ImageNet-100 between intra-class and inter-class samples. (c) The influence of mask ratio on the relative distance between intra-class and inter-class samples.

performance in MAE. In particular, we notice that the mask ratio $\rho$ has a decisive influence on both factors. On the one hand, the label error $\alpha$ will grow with a larger mask ratio, but it is only very large under an extremely high mask ratio. Intuitively, $\alpha$ represents the possibility of recovering the original class $y$ from the left patches after masking. As illustrated in Figure 2(a), small or medium mask ratio, even a large one of 0.75, hardly alters the belonging class, and only when the mask ratio is extremely high, *e.g.,* 0.95, we can hardly tell it is still a car. On the other hand, according to the spectral graph theory [7], the residual eigenvalues $\{\lambda_{k+1}, \ldots, \lambda_{N_1}\}$ represent the high frequency components of the graph, which have large magnitude when the graph is less connected, *e.g.,* having many disjoint components. Therefore, a small downstream error requires the augmentation graph $\mathcal{G}_A$ to have better connectivity. In this aspect, Figure 2(a) shows that when the mask ratio increases, many dissimilar patterns are masked out and the left patches can have an increasing level of similarity, particularly among intra-class samples, *e.g.,* the tires of the two cars. Therefore, a large mask ratio can effectively improve the connectivity of the augmentation graph by reducing sample variation and increasing inter-sample similarity. Considering both the effects on the label error $\alpha$ and the residual components $\{\lambda_{k+1}, \ldots, \lambda_{N_1}\}$, we notice that there is a tradeoff in the choice of mask ratio: we should choose a properly large one to increase graph connectivity, and avoid too large mask ratio as it leads to the class mixture. In other words, a guiding principle is that a mask ratio should be chosen such that the mask-induced graph connectivity should happen mostly among *intra-class* samples (no harm) instead of *inter-class* samples (inducing large label error). Below, we further examine this understanding on real-world data.

**Empirical Investigation.** To verify our perspective, we conduct experiments on CIFAR-10 with different mask ratios. We evaluate the distance between two images by calculating the average $l_2$ distance between every pair of patches. As shown in Figure 2(b), we find that the average distance of intra-class images decreases with the increasing of mask ratio, which means that a higher mask ratio leads to generating more edges between intra-class samples with better connectivity (*i.e.,* decreasing magnitude of residual eigenvalues $\{\lambda_{k+1}, \ldots, \lambda_{N_1}\}$). Meanwhile, we also notice that a high mask ratio will also lead to a smaller inter-class distance, which suggests an increasing label error $\alpha$. Therefore, there exists a tradeoff on the mask ratio to balance residual eigenvalues $\{\lambda_{k+1}, \ldots, \lambda_{N_1}\}$ and labeling error $\alpha$.

**The Sweet Spot for Mask Ratio.** The discussion above reveals that mask ratio has an effect on both intra-class and inter-class connectivity, and a good mask ratio should be selected to balance the two sides such that intra-class distance is relatively high while inter-class distance is relatively low. Motivated by this, we compute the relative distance (intra-class over inter-class). As shown in Figure 2(c), the relative distance decreases first with the increasing mask ratio, showing that the intra-class distance decreases faster than the inter-class distance under small mask ratio. When $\rho > 0.7$, the relative distance becomes larger again, indicating that the difference between intra-class and inter-class edges disappears under too large mask ratio. The sweet spot lies in $\rho = 0.7$, which is pretty close to the optimal mask ratio of MAE ($\rho = 0.75$). This shows that our theoretical analysis of the effect of mask ratio aligns surprisingly well with the practice of MAE.

Table 1: Linear evaluation accuracy (%) and fine-tuning accuracy (%) of pretrained models by MAE loss and U-MAE loss with different ViT backbones on CIFAR-10, ImageNet-100, and ImageNet-1K. The uniformity regularizer term in the U-MAE loss significantly improves the linear evaluation performance of the MAE loss without hurting the performance of fine-tuning accuracy.

| Downstream Task | Method | CIFAR-10 | | ImageNet-100 | | ImageNet-1K | |
| | | ViT-Tiny | ViT-Base | ViT-Base | ViT-Large | ViT-Base | ViT-Large |
|---|---|---|---|---|---|---|---|
| Linear Probing | MAE | 59.6 | 61.7 | 61.2 | 64.4 | 55.4 | 62.2 |
| | U-MAE | **68.9** | **70.2** | **67.5** | **72.8** | **58.5** | **65.8** |
| Fine-tuning | MAE | **89.6** | 90.7 | **86.9** | 87.3 | 82.9 | **83.3** |
| | U-MAE | 89.4 | **90.8** | 86.8 | 87.3 | **83.0** | 83.2 |

# 5 Experiments

In this section, we first present the main empirical results of our proposed U-MAE loss on different real-world datasets with different backbones. Then we conduct a series of experiments to understand how well the U-MAE loss works.

## 5.1 Evaluation on Benchmark Datasets

To evaluate the effectiveness of the proposed U-MAE loss, extensive experiments are conducted on CIFAR-10 [21], ImageNet-100 [8], and ImageNet-1K [8].

**Setup.** We mainly follow the basic setup of MAE [15]: for the encoder, we adopt different variants of ViT [10], *i.e.,* ViT-Tiny, ViT-Base, and ViT-Large. For the decoder, we use a flexible one following [15]. The mask ratio is set to 0.75. For U-MAE, the coefficient of the uniformity term is set to 0.01. On CIFAR-10, we pretrain the model for 2000 epochs with batch size 4096 and weight decay 0.05. On ImageNet-100 and ImageNet-1K, we pretrain the model for 200 epochs with batch size 1024 and weight decay 0.05. We conduct both linear evaluation and non-linear fine-tuning on the pretrained encoder. For linear evaluation, we train a linear classifier on the frozen pretrained encoder. As for non-linear fine-tuning, we train both the pretrained encoder and the linear classifier with the cross entropy loss.

**Effectiveness of the Proposed U-MAE Loss.** In Table 1, we compare the performance of original MAE loss and U-MAE loss on different benchmarks. We find that, on linear evaluation results, our proposed U-MAE loss increases 8.9% on CIFAR-10, 7.2 % on ImageNet-100, and 3.4% on ImageNet-1K with two different backbones. On fine-tuning results, our proposed U-MAE loss will not hurt the performance of fine-tuning results of MAE. The experimental results verify the effectiveness of the proposed U-MAE loss, which achieves better performance than the original MAE loss across different datasets and different backbones. Moreover, to quickly evaluate the training process of the encoder, we set an online linear classifier to monitor the linear accuracy whose results can be found in Appendix B.5.

**Extention to Other MIM Frameworks.** We also verify our proposed method in another MIM framework SimMIM [29]. For SimMIM, we use ViT-Base as the encoder and use the linear decoder as in [29]. We use the recommended mask ratio 0.6. Similar to U-MAE, Uniformity-enhanced SimMIM loss (U-SimMIM) adds a uniformity regularization term to the original reconstruction loss of SimMIM, where the coefficient of the uniformity term is set to 0.01. For ImageNet-100, we pretrain the model for 200 epochs with batch size 128 and weight decay 0.05. Linear evaluation is conducted in Table 2. We can see that the linear accuracy increases 6.8% with the uniformity regularizer, which further verifies the general property of our approach.

Table 2: Linear probing accuracy (%) of U-SimMIM (ViT-Base) on ImageNet-100.

| SimMIM | U-SimMIM |
|---|---|
| 54.3 | **61.1** |

Table 3: Linear probing accuracy (%) of U-MAE with different coefficients (Eq. (10)).

| $\lambda$ | 0 | 1e-3 | 1e-2 | 1e-1 | 1e0 |
|---|---|---|---|---|---|
| Acc | 61.2 | 65.9 | **67.5** | 45.1 | 47.0 |

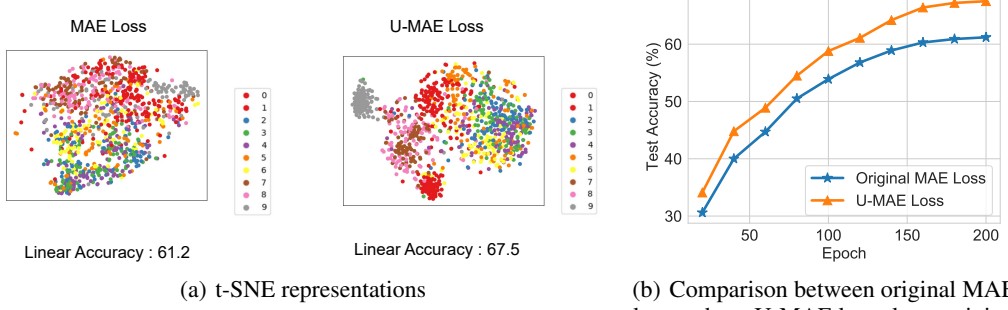

(a) t-SNE representations

(b) Comparison between original MAE loss and our U-MAE loss along training.

Figure 3: (a) Visualization of representations on random 10 classes of ImageNet-100 trained with MAE loss and our U-MAE loss. Our U-MAE loss significantly improves the class-clustering performance of the encoder. (b) The linear evaluation results during the training process. It shows that our U-MAE loss improves the downstream performance consistently along training.

## 5.2 Empirical Understandings

**Visualization of Representations.** To intuitively understand the improvement of our U-MAE loss on clustering intra-class samples, we use t-SNE [26] to visualize the representations trained with MAE loss and U-MAE loss on ten random class of ImageNet-100 (detailed classes are introduced in Appendix B.4). We find that with our uniformity regularizer term, the samples are much better-clustered corresponding to their ground-truth labels. To be specific, the red class ("hens") and the gray class ("indigo birds") are separated from others, this is because most of other classes are the animals living in the oceans while these two classes are more like the birds living on the land or the sky. Thus, these two classes are easier to be distinguished, especially with our uniformity regularizer.

**Different Coefficients of the Regularizer Term.** The most important hyper-parameter of our proposed U-MAE loss is the coefficient of the uniformity regularizer term. In Table 3, we present the results of linear evaluation on ImageNet-100 trained with the U-MAE loss with different coefficients of the regularizer term. We can see that the downstream performance increases when the coefficient increases from 0 to 0.01. However, the overlarge coefficient will also hurt the performance of U-MAE loss as the task of MAE will be overlooked.

**Training process.** To further compare the performance between the original MAE loss and U-MAE loss, we plot the linear evaluation accuracy on ImageNet-100 during the training process in Figure 3(b). We can observe that our proposed U-MAE loss improves the performance of MAE with all different training epochs, which verifies the effectiveness of our proposed U-MAE loss.

## 6 Conclusion

In this paper, we proposed a new theoretical understanding of MAE by formally connecting MAE and contrastive learning. In particular, we show that a small MAE loss implies a good alignment between the mask-induced positive samples. Based on this connection, we further analyze the dimensional collapse issue of MAE and propose a new variant, named Uniformity-enhanced MAE (U-MAE), through adding an explicit feature uniformity regularization. Theoretically, this connection enables us to establish theoretical guarantees on the downstream classification error, which also provides theoretical insights on the choice of large mask ratio in MAE. Empirically, the proposed U-MAE successfully mitigates the dimensional collapse issue of MAE, and achieves consistent improvement on the linear probing task on CIFAR-10, ImgaeNet-100, and ImageNet-1K.

## Acknowledgment

Yisen Wang is partially supported by the NSF China (No. 62006153), Project 2020BD006 supported by PKU-Baidu Fund, and Huawei Technologies Inc. We thank anonymous reviewers for constructive and helpful discussions.

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
