# A Proofs

## A.1 Proof of Theorem 3.4

*Proof.* With Assumption 3.1, we have

$$
\begin{aligned}
\mathcal{L}_{\text{MAE}}(h) &= \mathbb{E}_{x_1,x_2}\|h(x_1) - x_2\|^2 \\
&= \mathbb{E}_{x_1,x_2}\|h(x_1) - x_2\|^2 + \varepsilon - \varepsilon \\
&\geq \mathbb{E}_{x_1,x_2}\|h(x_1) - x_2\|^2 + \|x_2 - h_g(x_2)\|^2 - \varepsilon && (\|x_2 - h_g(x_2)\|^2 \leq \varepsilon) \\
&\geq \frac{1}{2}\mathbb{E}_{x_1,x_2}\|h(x_1) - h_g(x_2)\|^2 - \varepsilon && (\|a + b\|^2 \leq 2(\|a\|^2 + \|b\|^2)) \\
&= -\mathbb{E}_{x_1,x_2}h(x_1)^\top h_g(x_2) - \varepsilon + 1. && (h(x) \text{ and } h_g(x) \text{ are normalized})
\end{aligned}
$$

We formulate the features as two matrices $H, H_g$. We denote $H(x_1) = \sqrt{d_{x_1}}h(x_1)$ as the $x_1$-th column of the matrix $H$ and $H_g(x_2) = \sqrt{d_{x_2}}h(x_2)$ as the $x_2$-th row of the matrix $H_g$. As defined before, $(A_M)_{x_2,x_1}$ is the joint distribution of $x_1$ and $x_2$, *i.e.,* $(A_M)_{x_2,x_1} = w_{x_1,x_2}$. We denote the normalized form of $A_M$ as $\bar{A}_M$, *i.e.,* $(\bar{A}_M)_{x_2,x_1} = \frac{w_{x_1,x_2}}{\sqrt{d_{x_1}}\sqrt{d_{x_2}}}$. Then we can reformulate the reconstruction loss,

$$
\begin{aligned}
\mathcal{L}_{\text{MAE}}(h) &\geq -\sum_{x_1,x_2} \frac{w_{x_1,x_2}}{\sqrt{d_{x_1}d_{x_2}}}\sqrt{d_{x_1}}h(x_1) \cdot \sqrt{d_{x_2}}h_g(x_2) - \varepsilon + 1 \\
&= -\operatorname{tr}(\bar{A}_M H H_g^\top) - \varepsilon + 1 \\
&\geq -\frac{1}{2}(\|\bar{A}_M H\|^2 + \|H_g\|^2) - \varepsilon + 1.
\end{aligned}
$$

As the output of decoder is normalized, *i.e.,* $\|H_g\|^2 = 1$. we obtain

$$
\mathcal{L}_{\text{MAE}}(h) \geq -\frac{1}{2}(\operatorname{tr}(\bar{A}_M^\top \bar{A}_M H H^\top) + 1) - \varepsilon + 1 = -\frac{1}{2}\operatorname{tr}(\bar{A}_M^\top \bar{A}_M H H^\top) - \varepsilon + \frac{1}{2}. \tag{16}
$$

Then we element-wise compute $\bar{A}_M^\top \bar{A}_M$ and $HH^\top$, we have

$$
(\bar{A}_M^\top \bar{A}_M)_{x_1,x_1^+} = \sum_{x_2} \frac{w_{x_1,x_2}w_{x_1,x_2^+}}{d_{x_2}\sqrt{d_{x_1}d_{x_1^+}}}. \tag{17}
$$

$$
(HH^\top)_{x_1^+,x_1} = \sqrt{d_{x_1}d_{x_1^+}}h(x_1)^\top h(x_1^+). \tag{18}
$$

As the trace is the sum of the diagonal value of the matrix, we consider $x_1$-th diagonal value of $(\bar{A}_M^\top \bar{A}_M H H^\top)$, *i.e.,*

$$
(\bar{A}_M^\top \bar{A}_M H H^\top)_{x_1,x_1} = \sum_{x_1^+}(A_M^\top A_M)_{x_1,x_1^+}(HH^\top)_{x_1^+,x_1} = \sum_{x_1^+}\sum_{x_2} \frac{w_{x_1,x_2}w_{x_1^+,x_2}}{d_{x_2}}h(x_1)^\top h(x_1^+). \tag{19}
$$

With that, we can element-wise expand Eq. (16),

$$
\mathcal{L}_{\text{MAE}}(h) \geq -\frac{1}{2}\mathbb{E}_{x_1,x_1^+\sim\hat{p}(x,x^+)}h(x_1)^\top h(x_1^+) - \varepsilon + \frac{1}{2},
$$

where $\hat{p}(x_1, x_1^+) = \sum_{x_2} \frac{w_{x_1,x_2}w_{x_1^+,x_2}}{d_{x_2}}$. Similarly, we define a matrix $H_g$, where $(H_g)_{x_2} = \sqrt{d_{x_2}}h_g(x_2)$. We let $(\bar{A}_M)_{x_2,x_1} = \frac{w_{x_1,x_2}}{\sqrt{d_{x_1}}\sqrt{d_{x_2}}}$ and we obtain

$$
\begin{aligned}
\mathcal{L}_{\text{MAE}}(h) &\geq -\operatorname{tr}(HH_g^\top \bar{A}_M) - \varepsilon + 1 \\
&\geq -\frac{1}{2}(\|H\|^2 + \|\bar{A}_M^\top H_g\|^2) - \varepsilon + 1 && (\operatorname{tr}(AB) \leq \frac{1}{2}\|A\|^2 + \|B\|^2) \\
&= -\frac{1}{2}(\operatorname{tr}(\bar{A}_M \bar{A}_M^\top H_g H_g^\top) + 1) - \varepsilon + 1 && (H_g \text{ is normalized}) \\
&\geq -\frac{1}{2}\mathbb{E}_{x_2^+\sim\bar{p}(x_2,x_2^+)}h_g(x_2)^\top h_g(x_2^+) - \varepsilon + \frac{1}{2},
\end{aligned}
$$

where $\bar{p}(x_2, x_2^+) = \sum_{x_1} \frac{w_{x_1,x_2} w_{x_1,x_2^+}}{d_{x_1}}$. $\qquad\qquad\qquad\qquad\qquad\qquad\qquad\qquad\qquad\qquad\qquad$ $\square$

## A.2  Proof of Corollary 3.5

With Theorem 3.4, we know that

$$\mathcal{L}_{\text{MAE}}(h) \geq -\frac{1}{2}\mathbb{E}_{x_1,x_1^+\sim\hat{p}(x,x^+)} h(x_1)^\top h(x_1^+) - \varepsilon + \frac{1}{2}.$$

As the decoder is $L$-bi-Lipschitz, we obtain

$$\forall (x_1, x_2), 1/L \cdot \|x_1 - x_2\|^2 \leq \|g(x_1) - g(x_2)\|^2 \leq L \cdot \|x_1 - x_2\|^2. \qquad (20)$$

So,

$$
\begin{aligned}
\mathcal{L}_{\text{MAE}}(h) &\geq -\frac{1}{2}\mathbb{E}_{x_1,x_1^+\sim\hat{p}(x_1,x_1^+)} h(x_1)^\top h(x_1^+) - \varepsilon + \frac{1}{2} \\
&= \frac{1}{4}\mathbb{E}_{x_1,x_1^+\sim\hat{p}(x_1,x_1^+)} \|h(x_1) - h(x_1^+)\|^2 - \varepsilon && (h(x) \text{ is normalized}) \\
&= \frac{1}{4}\mathbb{E}_{x_1,x_1^+\sim\hat{p}(x_1,x_1^+)} \|g(f(x_1)) - g(f(x_1^+))\|^2 - \varepsilon \\
&\geq \frac{1}{4L}\mathbb{E}_{x_1,x_1^+\sim\hat{p}(x_1,x_1^+)} \|f(x_1) - (f(x_1^+)\|^2 - \varepsilon && (\text{Equation (20)}) \\
&= -\frac{1}{2L}\mathbb{E}_{x_1,x_1^+\sim\hat{p}(x_1,x_1^+)} f(x_1)^\top f(x_1^+) - \varepsilon + \frac{1}{2}.
\end{aligned}
$$

## A.3  Proof of Theorem 3.6

*Proof.* When the encoder fully collapses, the encoder $f$ maps all the features to the same point $c$, *i.e.,*

$$\forall x \in \mathcal{X}_1, f(x) = c. \qquad (21)$$

Then,

$$
\begin{aligned}
\mathcal{L}_{MAE}(h) &= \mathbb{E}_{x_1,x_2} \|g(f(x_1)) - x_2\|^2 \\
&= \mathbb{E}_{x_2} \|g(c) - x_2\|^2
\end{aligned}
\qquad (22)
$$

We then select a g(c) to make Equation (22) minimal. According to KKT conditions, it has a closed-form solution $q^\star$, satisfying

$$2\mathbb{E}_{x_2}(q^\star - x_2) = 0. \qquad (23)$$

*i.e.,* $Q^\star = \mathbb{E}_{x_2} x_2$. Then

$$
\begin{aligned}
\mathcal{L}_{MAE}(h) &= \mathbb{E}_{x_2} \|g(c) - x_2\|^2 \\
&\geq \mathbb{E}_{x_2} \|\mathbb{E}_{x_2'} x_2' - x_2\|^2 \\
&= \text{Var}(x_2).
\end{aligned}
$$

$\qquad\qquad\qquad\qquad\qquad\qquad\qquad\qquad\qquad\qquad\qquad\qquad\qquad\qquad\qquad\qquad$ $\square$

## A.4  Proof of Theorem 3.7

With Corollary 3.5, we have

$$\mathcal{L}_{\text{MAE}}(h) \geq -1/(2L) \cdot \mathbb{E}_{x_1,x_1^+} f(x_1)^\top f_g(x_1^+) - \varepsilon + const. \qquad (24)$$

Then we set $\lambda = \frac{1}{4L}$ and we obtain

$$
\begin{aligned}
\mathcal{L}_{\text{U-MAE}}(h) &= \mathcal{L}_{MAE}(h) + 1/(4L) \cdot \mathcal{L}_{unif}(f) \\
&\geq 1/(2L) \cdot \mathcal{L}_{align}(f) + 1/(4L) \cdot \mathcal{L}_{unif}(f) - \varepsilon + const \\
&= 1/(4L) \cdot \mathcal{L}_{SCL}(f) - \varepsilon + const.
\end{aligned}
\qquad (25)
$$

### A.5 Proof of Theorem 4.1

*Proof.* We compose the marginal distribution of $x_1$ as a matrix $D$ and $D_{x_1} = d_{x_1}$ is the $x_1$-th row of $D$. And we denote $U$ as the matrix composed of encoder features, *i.e.*, $U_{x_1} = \sqrt{d_{x_1}} f(x_1)$. Recall that $A_{x_1,x_1^+} = \mathcal{A}(x_1, x_1^+)$ and $\bar{A}$ is the normalized form of $A$, *i.e.*, $A_{x_1,x_1^+} = \frac{\mathcal{A}(x_1,x_1^+)}{\sqrt{d_{x_1} \cdot d_{x_1^+}}}$. Then we reformulate the downstream error,

$$
\begin{aligned}
\mathbb{E}_{x,y}\|y - W_f f(x)\|^2 &= \sum_{(x_1, y_{x_1})} d_{x_1} \|y_{x_1} - W_f f(x_1)\|^2 \\
&= \|D^{1/2}Y - UW_f\|^2 \\
&= \|D^{1/2}Y - \bar{A}C + \bar{A}C - UW_f\|^2,
\end{aligned}
$$

where $C_{x_1,j} = \sqrt{(d_i)\mathbb{1}_{y_{x_1}=j}}$. Then we consider the relationship between the downstream error and the augmentation graph, we element-wise consider the matrix $(D^{1/2}Y - \bar{A}C)$,

$$
(D^{1/2}Y)_{x_1,j} = \sqrt{(d_{x_1})\mathbb{1}_{y_{x_1}=j}}, \quad (\bar{A}C)_{x_1,j} = \sum_{x_1^+} \frac{w_{x_1,x_1^+}}{\sqrt{d_{x_1}} \cdot \sqrt{d_{x_1^+}}} \sqrt{(d_{x_1^+})\mathbb{1}_{y_{x_1^+}=j}}. \tag{26}
$$

So when $j = y_{x_1}$,

$$
\begin{aligned}
(D^{1/2}Y - \bar{A}C)_{x_1,j} &= \sqrt{(d_{x_1})\mathbb{1}_{y_{x_1}=j}} - \sum_{x_1^+} \frac{\mathcal{A}(x_1, x_1^+)}{\sqrt{d_{x_1}}} \mathbb{1}_{y_{x_1^+}=j} \\
&= \sqrt{d_{x_1}} - \sum_{x_1^+} \frac{\mathcal{A}(x_1, x_1^+)}{\sqrt{d_{x_1}}} \mathbb{1}_{y_{x_1^+}=j} \\
&= \sum_{x_1^+} \frac{\mathcal{A}(x_1, x_1^+)}{\sqrt{d_{x_1}}} - \sum_{x_1^+} \frac{\mathcal{A}(x_1, x_1^+)}{\sqrt{d_{x_1}}} \mathbb{1}_{y_{x_1^+}=j} \\
&= \sum_{x_1^+} \frac{\mathcal{A}(x_1, x_1^+)}{\sqrt{d_{x_1}}} \mathbb{1}_{y_{x_1^+} \neq j} \\
&= \sum_{x_1^+} \frac{\mathcal{A}(x_1, x_1^+)}{\sqrt{d_{x_1}}} \mathbb{1}_{y_{x_1^+} \neq y_{x_1}}.
\end{aligned} \tag{27}
$$

When $j \neq y_{x_1}$,

$$
\begin{aligned}
(D^{1/2}Y - \bar{A}C)_{x_1,j} &= \sqrt{(d_{x_1})\mathbb{1}_{y_{x_1}=j}} - \sum_{x_1^+} \frac{\mathcal{A}(x_1, x_1^+)}{\sqrt{d_{x_1}}} \mathbb{1}_{y_{x_1^+}=j} \\
&= 0 - \sum_{x_1^+} \frac{\mathcal{A}(x_1, x_1^+)}{\sqrt{d_{x_1}}} \mathbb{1}_{y_{x_1^+}=j} \\
&= -\sum_{x_1^+} \frac{\mathcal{A}(x_1, x_1^+)}{\sqrt{d_{x_1}}} \mathbb{1}_{y_{x_1^+}=j}.
\end{aligned} \tag{28}
$$

We define $\beta_{x_1} = \sum_{x_1^+} \mathcal{A}(x_1, x_1^+) \mathbb{1}_{y_{x_1^+} \neq y_{x_1}}$, and we have

$$
\begin{aligned}
\|(D^{1/2}Y - \bar{A}C)_{x_1}\|^2 &= \left(\sum_{x_1^+} \frac{\mathcal{A}(x_1, x_1^+)}{\sqrt{d_{x_1}}} \mathbb{1}_{y_{x_1^+} \neq y_{x_1}}\right)^2 + \sum_{j \neq y_{x_1}} \left(\sum_{x_1^+} \frac{\mathcal{A}(x_1, x_1^+)}{\sqrt{d_{x_1}}} \mathbb{1}_{y_{x_1^+} = j}\right)^2 \\
&\leq \left(\sum_{x_1^+} \frac{\mathcal{A}(x_1, x_1^+)}{\sqrt{d_{x_1}}} \mathbb{1}_{y_{x_1^+} \neq y_{x_1}}\right)^2 + \left(\sum_{j \neq y_{x_1}} \sum_{x_1^+} \frac{\mathcal{A}(x_1, x_1^+)}{\sqrt{d_{x_1}}} \mathbb{1}_{y_{x_1^+} = j}\right)^2 \\
&\leq \left(\sum_{x_1^+} \frac{\mathcal{A}(x_1, x_1^+)}{\sqrt{d_{x_1}}} \mathbb{1}_{y_{x_1^+} \neq y_{x_1}}\right)^2 + \left(\sum_{x_1^+} \frac{\mathcal{A}(x_1, x_1^+)}{\sqrt{d_{x_1}}} \sum_{j \neq y_{x_1}} \mathbb{1}_{y_{x_1^+} = j}\right)^2 \quad (29) \\
&= \left(\sum_{x_1^+} \frac{\mathcal{A}(x_1, x_1^+)}{\sqrt{d_{x_1}}} \mathbb{1}_{y_{x_1^+} \neq y_{x_1}}\right)^2 + \left(\sum_{x_1^+} \frac{\mathcal{A}(x_1, x_1^+)}{\sqrt{d_{x_1}}} \mathbb{1}_{y_{x_1^+} \neq y_{x_1}}\right)^2 \\
&= \frac{2\beta_{x_1}^2}{d_{x_1}}.
\end{aligned}
$$

With that, we obtain $\|D^{1/2}Y - \bar{A}C\| = \sum_{x_1} \frac{2\beta_{x_1}^2}{d_{x_1}}$. As we assume that $E_{\bar{x} \sim \mathcal{P}_d}(\mathcal{A}(x_1|\bar{x}) \mathbb{1}_{y_{x_1} \neq \bar{y}}) \leq \alpha$, so

$$
\begin{aligned}
\sum_{x_1, x_1^+} \mathcal{A}(x_1, x_1^+) \mathbb{1}_{y_{x_1^+} \neq y_{x_1}} &= \sum_{x_1, x_1^+} E_{\bar{x}}(\mathcal{A}(x_1|\bar{x}) \mathcal{A}(x_{x_1^+}|\bar{x}) \mathbb{1}_{y_{x_1^+} \neq y_{x_1}}) \\
&\leq \sum_{x_1, x_1^+} E_{\bar{x}}(\mathcal{A}(x_1|\bar{x}) \mathcal{A}(x_{x_1^+}|\bar{x})(\mathbb{1}_{y_i \neq \bar{y}} + \mathbb{1}_{y_{x_1^+} \neq \bar{y}})) \quad (30) \\
&= 2E_{\bar{x} \sim \mathcal{P}_d}(\mathcal{A}(x_1|\bar{x}) \mathbb{1}_{y_{x_1} \neq \bar{y}}) \\
&\leq 2\alpha.
\end{aligned}
$$

Then we have

$$
\begin{aligned}
\|D^{1/2}Y - \bar{A}C\| &= \sum_{x_1} \frac{2\beta_{x_1}^2}{d_{x_1}} \\
&\leq \sum_{x_1} 2\beta_{x_1} \qquad\qquad\qquad (d_{x_1} = \sum_{x_1^+} \mathcal{A}(x_1, x_1^+) \geq \sum_{x_1^+} \mathcal{A}(x_1, x_1^+) \mathbb{1}_{y_{x_1} \neq y_{x_1^+}}) \\
&= 2 \sum_{x_1, x_1^+} \mathcal{A}(x_1, x_1^+) \mathbb{1}_{y_{x_1} \neq y_{x_1^+}} \qquad\qquad\qquad \text{(definition of } \beta_{x_1}) \\
&\leq 4\alpha. \qquad\qquad\qquad\qquad\qquad\qquad \text{(Equation (30))}
\end{aligned}
$$

Then we obtain

$$
\begin{aligned}
&\mathbb{E}_{x,y}\|y - W_f f(x)\|^2 \\
&= \|D^{1/2}Y - \bar{A}C + \bar{A}C - UW_f\|^2 \\
&\leq 2\|\bar{A}C - UW_f\|^2 + 8\alpha \qquad\qquad\qquad\qquad (\|A+B\|^2 \leq \|A\|^2 + \|B^2\|) \\
&= 2\|(\bar{A} - UU^T + UU^T)C - UW_f)\|^2 + 8\alpha \\
&= 2\|(\bar{A} - UU^T)C + U(U^TC - W_f)\|^2 + 8\alpha \\
&\leq 4(\|(\bar{A} - UU^T)C\|^2 + \|U(U^TC - W_f)\|)^2 + 8\alpha \qquad (\|A+B\|^2 \leq 2(\|A\|^2 + \|B\|^2)) \\
&\leq 4(\|(\bar{A} - UU^T)\|^2\|C\|^2 + \|U\|^2\|(U^TC - W_f)\|^2) + 8\alpha \qquad (\|AB\| \leq \|A\|\|B\|) \\
&\leq 4(\|(\bar{A} - UU^T)\|^2 + \|(U^TC - W_f)\|^2) + 8\alpha \qquad\qquad (\|C\| = 1, \|U\| = 1) \\
&= 4\mathcal{L}_{SCL}(f) + const + \|(U^TC - W_f)\|^2 + 8\alpha \qquad\qquad\qquad \text{(Lemma B.8 in [13])} \\
&\leq 16L \cdot \mathcal{L}_{\text{U-MAE}}(h) + 8\mathbb{E}_y[\mathbb{E}_{x|y}f(x) - b_y]^2 + 8\alpha + 16L\varepsilon + const. \qquad \text{(Theorem 3.7)} \\
&\leq 16L \cdot \mathcal{L}_{\text{U-MAE}}(h) + 8\alpha + 16L\varepsilon + const. \qquad\qquad\qquad (W_f \text{ is a mean classifier)} \\
&\qquad\qquad\qquad\qquad\qquad\qquad\qquad\qquad\qquad\qquad\qquad\qquad\qquad\qquad\qquad\qquad\qquad (31)
\end{aligned}
$$

In the next step, we analyze the prediction error. We denote $\bar{y}$ as the ground-truth label of original data $\bar{x}$. We first define a ensembled linear predictor $c'_f$. For an original sample, the predictor ensembles the results of all different views and choose the label predicted most. With the definition, $\bar{y} \neq c'_f(\bar{x})$ only happens when more than half of the views predict wrong labels. So

$$
\begin{aligned}
&\Pr(\bar{y} \neq c'_f(\bar{x})) \\
&\leq 2\Pr(\bar{y} \neq c_f(x)) \\
&\leq 4\mathbb{E}_{\bar{x} \sim \mathcal{P}_d(x), x \sim \mathcal{M}_1(x|\bar{x})} \|\bar{y} - W_f f(x)\|^2 && \text{(Claim B.9 in [13])} \\
&\leq 8(\mathbb{E}_{x,y}\|y - W_f f(x)\|^2 + \mathbb{E}_{\bar{x} \sim \mathcal{P}_d(x), x \sim \mathcal{M}_1(x|\bar{x})}\|y - \bar{y}\|^2) && (\|A + B\|^2 \leq \|A\|^2 + \|B^2\|) \\
&\leq 8(\mathbb{E}_{x,y}\|y - W_f f(x)\|^2 + 2\alpha) && \text{(definition of } \alpha) \\
&\leq 32\mathcal{L}_{SCL}(f) + 64\alpha + 16\alpha + const \\
&\leq 128L \cdot \mathcal{L}_{\text{U-MAE}}(h) + 64\alpha + 16\alpha + 128L\varepsilon + const \\
&\leq 128L \cdot \mathcal{L}_{\text{U-MAE}}(h) + 80\alpha + 128L\varepsilon + const.
\end{aligned}
$$

$\square$

## A.6  Proof of Theorem 4.2

*Proof.* With Equation (31), we have

$$
\mathcal{L}_{\text{U-MAE}}(h) \geq \frac{1}{4L}\|A - UU^\top\|^2 - \varepsilon + const. \tag{32}
$$

We set $\mathcal{L}_{mf}(U) = \|(\bar{A} - UU^T)\|^2$. When $U^\star$ is the minimizer of $\mathcal{L}_{mf}(U)$, according to the analysis in [11], we obtain

$$
\|(\bar{A} - U^\star(U^\star)^T)\| = \sum_{i=k+1}^{N_1} \lambda_i^2, \tag{33}
$$

where $\lambda_{k+1} \cdots \lambda_{N_1}$ are the $N_1 - k$ largest eigenvalues of matrix $\bar{A}$. We denote $h^\star$ is the minimizer of $\mathcal{L}_{\text{U-MAE}}$ and $f^\star$ is the corresponding encoder. Then $U_{f^\star}$ is composed of the features encoded by $f^\star$, i.e., $(U_{f^\star})_{x_1} = \sqrt{d_{x_1}} f^\star(x_1)$. So for all $h \in \mathcal{H}$, we have

$$
\begin{aligned}
\mathcal{L}_{\text{U-MAE}}(h) \geq \mathcal{L}_{\text{U-MAE}}(h^\star) &\geq \frac{1}{4L}\|A - U_{f^\star}(U_{f^\star})^\top\|^2 - \varepsilon + const \\
&\geq \frac{1}{4L}\|A - U^\star(U^\star)^\top\|^2 - \varepsilon + const \\
&= \frac{1}{4L}(\sum_{i=k+1}^{N_1} \lambda_i)^2 - \varepsilon + const.
\end{aligned} \tag{34}
$$

$\square$

# B  Additional Experiment

## B.1  Experiment Details of Computing Effective Rank

We conduct experiments on ImageNet-100 and use ViT-Base as the backbone. We compare two kinds of pretrained encoders: 1) the encoder trained with original MAE loss, 2) the encoder trained with U-MAE loss ($\lambda = 0.0001$). Then we store the normalized encoded features. We construct a feature matrix $A$ with $n$ (n=100) random samples and compute its singular values $\sigma_1, \ldots, \sigma_n$. Then, we compute the effective rank [24] of the feature matrix as follows. We first compute the distribution of sigular values, i.e., $p_k = \frac{\sigma_k}{\|\sigma\|_1}$. Then we can obtain the effective rank with that:

$Erank(A) = \exp(-\sum_{k=1}^{n} p_k \log(p_k)).$

Table 4: Online linear accuracy of MAE and U-MAE loss.

| Method | CIFAR-10 | | ImageNet-100 | | ImageNet-1K | |
|--------|----------|----------|-----------|-----------|-----------|-----------|
| | ViT-Tiny | ViT-Base | ViT-Base | ViT-Large | ViT-Base | ViT-Large |
| MAE | 52.9 | 59.5 | 37.5 | 39.5 | 39.7 | 43.4 |
| U-MAE | **69.4** | **72.0** | **56.3** | **61.4** | **46.5** | **52.1** |

## B.2 Experiment Details of Section 4.2

We conduct the verification experiment of Section 4.2 on ImageNet-100. We set patch size to 16 and use random masking. When computing the distance, we compute the max $l_2$ distance between the patches of two images and denote it as the distance between two images. Then we compute the average distance of different images. When computing the distance of intra-class samples, we only compute the distance between the samples in the same class. While for inter-class distance, we only compute the distance between the samples of different classes. To reduce the amount of calculation, we random select 10 classes of ImageNet-100 for this experiment.

## B.3 Additional Experiment Details of Section 5.1

For CIFAR-10, we train ViT-Tiny on $1\times$NVIDIA V100 with 1600 epochs, which needs about 23 hours, and ViT-Base on $1\times$NVIDIA V100 with 1600 epochs, which needs about 25 hours. For ImageNet-100, we train ViT-Base on $4\times$NVIDIA V100 with 200 epochs, which needs about 37 hours, and ViT-Large on $4\times$NVIDIA V100 with 200 epochs, which needs about 42 hours.

## B.4 Details of Visualization Results in Figure 3(a)

We use t-SNE to to visualize the representations learned with MAE and U-MAE loss on random 10 classes of ImageNet. The ten classes are 0) "cock", 1) "hen", 2) "tiger shark, Galeocerdo cuvieri", 3) "tench, Tinca tinca", 4) "goldfish, Carassius auratus", 5) "hammerhead, hammerhead shark", 6) "electric ray, crampfish, numbfish, torpedo", 7) "stingray", 8) "great white shark, white shark, man-eater, man-eating shark, Carcharodon carcharias", 9) "indigo bunting, indigo finch, indigo bird, Passerina cyanea",

## B.5 Online Linear Evaluation Experiments

Besides the offline linear evaluation results in Section 5, we also present the results obtained by an online linear classifier. Specifically, we train a linear head along the MAE training process and detach its gradients. From Table 4, we find that our promoting loss increases 14.71% for linear evaluation results on CIFAR-10 with two different backbones and increases 7.35 % on ImageNet-100 with two different backbones. And we could see that our U-MAE also significantly outperforms MAE on large-scale datasets by improving 6.77% Top-1 accuracy on ImageNet-1K.

## B.6 Additional Experiments on SimMIM

For SimMIM, we use ViT-Base as the encoder and use the linear decoder as proposed in SimMIM [29]. We use the recommended mask ratio, 60%. Similar to MAE, we propose a Uniformity-promoting SimMIM loss (U-SimMIM) which adds a uniformity regularizer term to the original reconstruction loss of SimMIM. For the uniformity term of our proposed loss, we set the coefficient of the uniformity term to 0.01. For ImageNet-100, we pretrain the model for 200 epochs with batch size 128 and weight decay 0.05. We conduct linear evaluation on the unsupervised pretrained encoder. From Table 5, we could see that our U-SimMIM also significantly outperforms SimMIM on large-scale datasets by improving 8.46% Top-1 accuracy on ImageNet-1K.

# C Extension to Other MIM Methods

The basic paradigm of current MIM frameworks is to reconstruct the masked patches from unmasked ones, while their differences mainly exist in the implementation details [1, 32, 29, 15], *e.g.,* 1) the

Table 5: Online linear accuracy of SimMIM and U-SimMIM loss on ImageNet-100 with ViT-Base.

| Method | Top-1 Accuracy (%) |
|--------|--------------------|
| SimMIM | 21.59% |
| **U-SimMIM** | **30.05%(+8.46%)** |

input of encoder: including masked patches (SimMIM, iBOT, BEiT) or not (MAE); 2) the decoder: one-layer (SimMIM, iBOT) or multi-layer (MAE, BEiT); and 3) the target: RGB (SimMIM, MAE) or tokenized discrete tokens (iBOT, BEiT). In fact, our theoretical framework is quite general, and can be easily extended to these variants:

- **Input of Encoder.** If masked patches are adopted, the input of encoder could be modeled as $\tilde{x}_1 = (x_1, p_{x_1+x_2})$ and $\tilde{x}_2 = (x_2, p_{x_1+x_2})$, where $p$ represents the position embeddings. In this case, both of them have the access to the entire position encoding $p_{x_1+x_2}$.

- **Decoder.** One-layer decoder is a special case of our multi-layer formulation. Moreover, the one-layer decoder obtains the invertibility and our analysis can be simplified.

- **Target.** The tokenized target only corresponds to a specific format of $x_2$, which does not affect our analysis of their alignment property.