# OpenReview forum: "How Mask Matters: Towards Theoretical Understandings of Masked Autoencoders"
_NeurIPS.cc/2022/Conference — NeurIPS 2022 Accept_

### Official Review · Reviewer_wuSY · 2022-07-09

**Rating:** 8
**Confidence:** 5
**Soundness:** 4 excellent
**Presentation:** 3 good
**Contribution:** 4 excellent

**Summary:**

The authors propose a new theoretical understanding of Masked Autoencoders (MAE). From the theoretical perspective, this paper is interesting and novel. I agree with their contributions as stated in the paper.

1. They prove that MAE loss is upper bounded by an implicit alignment loss and develop a new understanding of how MAE bridges the samples in the same class. Based on their analysis, they propose a new U-MAE loss.

2. They establish a guarantee for the downstream performance of MAE with practical assumptions and compare the training process of contrastive learning and MAE.

3. They verify that their U-MAE loss can significantly improve the performance of MAE on two datasets CIFAR-10 and ImaegNet-100.

 As a very early attempt, this paper provides theoretical contributions for the MAE study. Both the "How to Understand Masked Autoencoders" paper and this paper provide insights for the theoretical research for the MAE community.

**Questions:**

Suggestions can be found in the above.

**Limitations:**

Training MAE from scratch is very resource-consuming. But for future work, it is still better to do experiments on the large-scale datasets, like ImageNet-1K used in the MAE paper.

**Strengths And Weaknesses:**

Strengths

MAE looks like a very straightforward model. But how to extract the mathematical essence of the simple model is very challenging. This paper contributes a novel understanding of MAE, and provides the experiments.   In particular, the authors propose a new theoretical understanding of why the choice of mask ratio is so important for MAE from a graph perspective. This is very insightful and inspiring.

Writing is clear and easy to follow.


Weaknesses -- some suggestions and minor issues.

1. The mathematical formulations can be improved.

For example, Line 95 [k] and Line 105 [.] will make the readers confused about the brackets.

In Line 100, the tilde can be replaced as $\in$, as D_u is defined as a set, not a distribution.

In Eq. (1), x_1 is defined as a vector, so it is better to use bold to denote the vector.

2. More implementation details.
It is very interesting to see more training details, including how many GPUs, and how many hours for training.

3. More explanation on experimental results.

For example, in Figure 4 (2), U-MAE loss obtained better visualizations. There are two clusters (grey and red) that are separated from other clusters. Why are they so distinctive from the others?  What are the 10 random classes?  What classes are the grey and red clusters presenting? What are the natures of these two classes in the original data domain?  Further analysis can give us more motivation.

---

> ### Author Response · Authors · 2022-08-02
> **Response to Reviewer wuSY**
>
>
> We thank Reviewer wuSY for appreciating our work on its novelty and insights, and highlighting its value to the field. Below, we address your main concerns.
>
> ---
> **Q1.** The mathematical formulations need to be improved.
>
> Thanks for pointing it out. We have revised the mathematical formulations following your advice.
>
>
> ---
>
> **Q2.** More implementation details. It is very interesting to see more training details, including how many GPUs, and how many hours for training.
>
> **A2.** Following your advice, we introduce more implementation details in **Appendix B.3**.
> In terms of the computation device, for CIFAR-10, we train ViT-Tiny on $1\times$NVIDIA V100 with 1600 epochs, which needs about 23 hours, and ViT-Base on $1\times$NVIDIA V100 with 1600 epochs, which needs about 25 hours. For ImageNet-100, we train ViT-Base on $4\times$NVIDIA V100 with 200 epochs, which needs about 37 hours, and ViT-Large on $4\times$NVIDIA V100 with 200 epochs, which needs about 42 hours.
>
> ---
>
> **Q3.** More explanations on experimental results.
> > For example, in Figure 4 (2), U-MAE loss obtained better visualizations. There are two clusters (gray and red) that are separated from other clusters. Why are they so distinctive from the others? What are the 10 random classes? What classes are the gray and red clusters presenting? What are the natures of these two classes in the original data domain? Further analysis can give us more motivation.
>
> **A3.** Thanks for your suggestions and we have added more details and analysis on our empirical results in Sec 5.2. Specifically, we have covered the problems you mentioned:
> - In Figure 4, the ten randomly selected classes are: 0) "cock", 1) "hen", 2) "tiger shark, Galeocerdo cuvieri", 3) "tench, Tinca tinca", 4) "goldfish, Carassius auratus", 5) "hammerhead, hammerhead shark", 6) "electric ray, crampfish, numbfish, torpedo", 7) "stingray", 8) "great white shark, white shark, man-eater, man-eating shark, Carcharodon carcharias", 9) "indigo bunting, indigo finch, indigo bird, Passerina cyanea". And we add captions to the figure.
> - Specifically, the red and gray classes are "hen" (1) and "indigo bunting, indigo finch, indigo bird, Passerina cyanea" (9), one on land and one in the sky, while many of the other eight classes are creators living in the ocean. Thus, the red and gray classes are easier to be separated from others, particularly with our uniformity regularizer.
>
> ---
>
> **Q4.** Training MAE from scratch is very resource-consuming. But for future work, it is still better to do experiments on the large-scale datasets, like ImageNet-1K used in the MAE paper.
>
> **A4.** Following your suggestion, we add experiments on **ImageNet-1k**. Due to the limit of time, we train **ViT-Base for 200 epochs** with MAE and UMAE objectives, respectively, and compare their linear probing accuracy to examine feature clustering quality. We adopt the official code of MAE, and for U-MAE, we simply adopt the default hyperparameters without tuning. Results are listed below.
>
> |Method | Top-1 Accuracy (%)  | Top-5 Accuracy (%) |
> |---|---|---|
> |MAE|39.72% | 65.62%
> |**U-MAE (ours)**| **46.49% (+6.77%)** |  **71.92% (+6.3%)** |
>
> We could see that our U-MAE also significantly outperforms MAE on large-scale datasets by improving 6.77% Top-1 and 6.3% Top-5 accuracy on ImageNet-1k. We haved added this result in **Appendix B.5**.
>
> ---
> Hope our revision and newly added experiment results could address your concerns. We are looking forward to your reply and please let us know if there is more to clarify.

---

> > ### Comment · Reviewer_wuSY · 2022-08-06
> > **post-rebuttal comments**
> >
> > Thanks for your detailed response. **I will keep my original rating.**
> >
> > This rebuttal is detailed and solid. It's great to see the new results on a larger dataset. The author discussed the transferability of their theory to other MIM methods. This paper is a good attempt and practice to study the MAE from a theoretical perspective.
> >
> > I've read the comments of all the reviewers.  **I'm happy to discuss more details with both reviewers and ACs.**

---

> > > ### Author Response · Authors · 2022-08-09
> > > **Thanks for your kind reply**
> > >
> > > Dear Reviewer wuSY,
> > >
> > > Thanks a lot for acknowledging our efforts and our novelty! We are so delighted to hear that you find our large-scale experiments and discussions with other MIM methods satisfactory. Please let us know if you have further concerns.
> > >
> > >
> > > Sincerely,
> > >
> > > Authors

---

### Official Review · Reviewer_qc3H · 2022-07-10

**Rating:** 4
**Confidence:** 4
**Soundness:** 2 fair
**Presentation:** 2 fair
**Contribution:** 2 fair

**Summary:**

The paper aims to provide a theoretical underpinning to explain the success of masked autoencoders (MAEs) in computer vision, with a particular focus on the importance of the mask ratio. To this end, the authors establish a connection between MAE training and contrastive self-supervised learning objectives, and leverage graph theory to establish a generalization bound. Furthermore, based on their analysis the authors propose a regularizer and test its effectiveness empirically, observing benefits on multiple data sets.

**Questions:**

- What is $\tilde x_2^-$ in equation (9)? How does the last term relate to uniformity as suggested by the terminology uniformity-promoting MAE? I could see how it relates to alignment.
- Why is (5) greater or equal the preceding expression? It seems it would be equal to the preceding expression.
- How are the graph weights define exactly? $M_1$ and $M_2$n are probability distributions if I read the definitions correctly, so the definition in (10) wouldn’t resolve to a scalar weight.
- What visualization technique is used to generate the plots in Figure 4? What data set are the results in Figure 5 obtained for?


**Limitations:**

It is unclear whether the approach really captures the mechanism underlying the success of MAE. At least I’m not convinced that the potentially incorrect derivations give additional insights on top of the intuitions e.g. about masking ratios given in the original MAE paper.

**Strengths And Weaknesses:**

I believe the search for a connection between contrastive self-supervised learning and reconstruction based approaches like MAEs is interesting and could turn out fruitful. The attempt to link the Siamese structure in contrastive SSL with the encoder-decoder architecture used in MAEs looks promising at first. However, I have doubts about different elements of the derivation of the alignment, as well as the presentation and the experiments.

The major issues I found are the following:
- It seems that then MAE encoder and decoder are incorretly defined in L93: Using the notation from the laper, the encoder $f$ maps $|m|$ image patches to $|m|$ tokens, which are then interleaved with $n - |m|$ mask tokens according to the locations of the masked patches and augmented with positional encodings. The resulting n tokens are then fed to the decoder $g$ which outputs $n$ patches. The authors then define $x_1$ corresponding to the stacked unmasked patches, and $x_2$ corresponding to the stacked masked masked patches. This means that $x_1$ and $x_2$ have different shapes unless the masking ratio is 0.5. As a consequence, the Lipschitz property cannot be applied as in L151 because $f(x_1, p_{x_1})$ and $f_g(x_2, p_{x_2})$ do not map into the same space. This would mean that the proof breaks down at this point. Interleaving $f$ and $f_g$ with the masked and unmasked tokens would bring them to comparable spaces, but considering the distance between the interleaved representations would not make much sense as this would amount to simply measuring the magnitude of the unmasked representation. I’m happy to change my assessment, but I don’t see how this can be fixed.
- Concepts from prior work are often unsufficiently explained, e.g. the derivation of the spectral graph theory based generalization bound and the related quantities (e.g. the authors talk about why the graph they consider is directed before it is clear what the relevance of the graph is). Notations are often defined after they are used for the first time, others are not introduced at all such as some quantities in Theorems 4.1 and 4.2.
- It is often unclear what the assumptions are and what is proven by theory. I believe a better way to structure the paper would be to clearly state the assumptions and then why they should be satisfied in practice (e.g. w.r.t. the Lipschitz assumption).


Minor weaknesses:
- L163 (and later) “regularizer” instead of “regular”
- The last parenthesis in (5) should be deleted.
- The choice of $f_g$ is important for the proposed regularization loss, but how it is chosen is only described in the appendix. It would be better to move the description to the main paper.

---

> ### Author Response · Authors · 2022-08-02
> **Response to Reviewer qc3H**
>
>
> We thank Reviewer qc3H for the detailed review and constructive comments. We have fixed the minor bugs and typos following your suggestions. Below we address your main concerns.
>
> ---
>
> **Q1.** It seems that the MAE encoder and decoder are incorrectly defined in L93:  $f$ and $f_g$ do not map into the same feature space.
>
>
> **A1.** We are afraid that there are some misunderstandings here. $f$ and $f_g$ are indeed mapping into the same feature space. To be specific, the encoder $f$ is a mapping that takes $x_1$ and $p_{x_1}$ ($|m|$ patches) as inputs, and outputs a **feature map of $|m|$ patches**, denoted as $z_1\in\mathbb{R}^{|m|\times q}$. Afterwards, the decoder $g$ maps $z_1$ and the position encoding $p_{x_2}\in\mathbb{R}^{(n-|m|)\times s}$ to the output space, denoted as $\hat{x}_2\in\mathbb{R}^{(n-|m|)\times p}$. Thus, to inverse the decoder mapping, the pseudo-inverse $f_g$ (defined in Assumption 3.2) is a mapping from $\hat{x}_2$ and $p(x_2)$ (with $n-|m|$ patches) to $z_1$ (with $|m|$ patches). In other words, the pseudo-inverse $f_g$ maps $x_2$ to the feature space of $x_1$. Formally, we have $f_g:\mathbb{R}^{(n-|m|)\times p}\times\mathbb{R}^{(n-|m|)\times s}\to\mathbb{R}^{|m|\times q}$. Thus, the output of $f_g$ indeed lies in the same output space as the encoder $f$, i.e. $\mathbb{R}^{|m|\times q}$, **even when the mask ratio $\rho\neq0.5$**. But indeed, our notations are not clear enough, and we have now revised them and added more clarifications on this point in **Sec 3.1 & 3.2**.
>
> As the matching of feature space appears to be a major concern of the reviewer, we respectfully ask the reviewer to consider reevaluating the paper if they are satisfied with the explanations above.
>
> ---
> **Q2.** Concepts from prior work are often insufficiently explained, e.g. the derivation of the spectral graph theory based generalization bound and the related quantities (e.g. the authors talk about why the graph they consider is directed before it is clear what the relevance of the graph is).
>
> **A2.** Thanks for pointing it out. We have re-organized our writing and added introductions to basic concepts and settings. Specifically, in Sec 4.1, we first establish the augmentation graph framework and introduce the basic concepts, and afterwards, we start to discuss the effect of directed graphs. We will continue improving our writing in the revision.
>
> ---
> **Q3.** It is often unclear what the assumptions are and what is proven by theory. I believe a better way to structure the paper would be to clearly state the assumptions and then why they should be satisfied in practice (e.g. w.r.t. the Lipschitz assumption).
>
> **A3.** Following your advice, we have re-organized our theoretical results in **Sec 4.1**, adding more clear discussions about our assumptions and the theoretical results.
> Specifically, we add a thorough explanation on what is proven by Theorem 4.2 in Sec 4.1, including the specific roles of $\alpha,L,\lambda_{k+1},\varepsilon$. We refer to the revised paper for more detailed discussions.
>
> ---
> **Q4.** The choice of $f_g$ is important for the proposed regularization loss, but how it is chosen is only described in the appendix. It would be better to move the description to the main paper.
>
> Thanks for pointing it out. Following your suggestion, we have moved the description on the choice of $f_g$ to the main paper (**Sec 3.2**) with more detailed elaborations.
>
> ---
> **Q5.** What is $x_2^-$ in Equation (9) ?
>
> **A5.** $x_2^-$ here refers to the negative sample, which is a random masked view of **an independently drawn sample**. We have added more details explaining U-MAE loss in **Sec 3.2**.
>
> ---
> **Q6.** Why is (5) greater or equal to the preceding expression?
>
> **A6.** Thanks for pointing it out. This is a typo and we have fixed it.
>
> ---
> **Q7.** How are the graph weights defined exactly?  $M_1$ and $M_2$ are probability distributions if I read the definitions correctly, so the definition in (10) wouldn’t resolve to a scalar weight.
>
> **A7.** In Eq (10), $M_1$ and $M_2$ actually refer to the probability density function (p.d.f.) of the augmentation distributions. So both $M_1(x|\bar x)$ and  $M_2(x'|\bar x)$ are scalars, and the expectation $w_{xx'}$ is also a scalar. We have revised our writing to clarify it in the revision (**Sec 3.1**).
>
> ---
> **Q8.**   What visualization technique is used to generate the plots in Figure 4? What dataset are the results in Figure 5 obtained for?
>
> **A8.**  The visualization technique used in Figure 4 is **t-SNE** [1] and the dataset used in Figure 5 is **ImageNet-100**. We have added these details in **Sec 5.2**.
>
> [1] Van der Maaten, Laurens, and Geoffrey Hinton. "Visualizing data using t-SNE." Journal of machine learning research 9.11 (2008).
>
>
> ---
>
> Hope our explanations above could address your concerns. We are looking forward to your reply, and please let us know if there is more to clarify.

---

> ### Author Response · Authors · 2022-08-06
> **Need further clarification?**
>
> Thanks very much for your constructive and detailed comments. We have tried our best to address the concerns and revised our paper accordingly. Is there any unclear point that we should/could further clarify?

---

> ### Author Response · Authors · 2022-08-07
> **Comments on the rebuttal?**
>
> Dear Reviewer qc3H,
>
> We understand that perhaps you are too busy to read the rebuttal. But since there are only two days left, we are sorry to remind you again.
>
> In our response, we have addressed your core concern on whether $f$ and $f_g$ map into the same feature space. For your concerns on the clarity of presentation, we have revised the paper organization and added more elaborations on theoretical results. We have also added new experiments on large scale datasets, ImageNet-1k, where our method U-MAE and U-SimMIM variant show significant improvement over the baselines.
>
> We are wondering if our response satisfies you? We are happy to answer any further questions.
>
> Sincerely,
>
> Authors

---

> ### Comment · Reviewer_qc3H · 2022-08-07
> **Response to author response**
>
> I would like to thank the authors for their clarification of the theory and the additional experiments. The derivations are now easier to read and better to understand.
>
> However, I’m still not convinced about the alignment hypothesis brought forward by the authors to link MAEs and contrastive learning - one of the major potential contributions in my opinion - for the following reasons. First, the reconstruction loss is upper-bounded by an alignment term (among other terms), which does not effectively establish that the two objectives induce similar representations. Second, the second encoder in the alignment term is one of potentially infinitely many pseudo inverse decoders, and not learned via a contrastive loss, so there is no indication that it will have similar properties to an encoder actually learned using a contrastive objective. Finally, the alignment term on the RHS of the last inequality in (5) (latest revision) vanishes if one assumes the decoder is invertible on its output space (such that $g(f_g(x_2, p_{x_2}), p_{x_2})=x_2$) which is not necessarily an unrealistic property.
>
> While I appreciate the additional ImageNet-1k experiments, I feel the baseline accuracy is low (I acknowledge the 200 epoch schedule).

---

> > ### Author Response · Authors · 2022-08-08
> > **Further Response to Reviewer qc3H (2/2)**
> >
> >
> > **Q2.** While I appreciate the additional ImageNet-1k experiments, I feel the baseline accuracy is low (I acknowledge the 200 epoch schedule).
> >
> > **A2.** Thanks for acknowledging our efforts. For the experiments on ImageNet-1k, we follow the **official code** of MAE and adopt default settings like the learning rate, weight decay and mask ratio for a fair reproduction. The results are summarized below.
> >
> > |Method | Top-1 Accuracy (%)  | Top-5 Accuracy (%) |
> > |---|---|---|
> > |MAE|39.72% | 65.62%
> > |**U-MAE (ours)**| **46.49% (+6.77%)** |  **71.92% (+6.3%)** |
> >
> > We can see that the improvement of our U-MAE is very significant (+6.77% top-1 acc) when we run both methods for 200 epochs. We will definitely provide more complete results in the revision.
> >
> >
> > ---
> > Thanks for your detailed reply and hope our explanations above could address your further concerns. We are looking forward to your further response, and please let us know if there is more to clarify.

---

> > ### Author Response · Authors · 2022-08-08
> > **Further Response to Reviewer qc3H (1/2)**
> >
> > Thanks for your reply! We are glad to hear that our detailed response has addressed your previous concerns. Below, we address your further concerns on the alignment analysis and the ImageNet-1k experiments.
> >
> > ---
> >
> > **Q1.** Concerns about the alignment hypothesis to link MAE and contrastive learning.
> >
> > **A1.** We will address your concerns point by point as follows.
> >
> > **Point a**
> > > First, the reconstruction loss is upper-bounded by an alignment term (among other terms), which does not effectively establish that the two objectives induce similar representations.
> >
> > Indeed, MAE's reconstruction loss is not equivalent to the alignment loss. Nevertheless, as MAE is rather new to us, it would be helpful to understand how it works by establishing connections to well studied methods like contrastive learning.
> > In deep learning literature, it is common to establish links between known and unknown objectives as a first step to grasp some insights. For example, in the seminal work of Arora et al [1] that first theoretically analyses contrastive learning, they also **upper bound** the contrastive learning loss with a corresponding supervised loss, and they analyze how contrastive learning works by studying each term in the upper bound. Our analysis also works in a similar fashion, by drawing some insights by studying the property of the upper bound. Below, we elaborate their connections in details.
> >
> > Theoretically, the two objectives are indeed deeply connected and induce similar features. Specifically, we added a new discussion in **Sec C.2**, and we show that if we further assume the decoder to be bi-Lipchitz, the implicit alignment loss is also a **lower bound** of MAE loss. As a result, a small MAE loss will also bring a smaller implicit alignment loss. In other words, ***a smaller reconstruction error implies a better alignment of positive views***. Therefore, the two methods are indeed theoretically related and their representations share similar properties like positive alignment.
> >
> >
> > Empirically, there are also many recent progresses (to list a few, [2,3,4]) showing that contrastive learning with random masking also achieves similar performance to MAEs, indicating that MAE indeed bears some inherent similarities to contrastive learning.
> >
> > [1] Arora et al., A theoretical analysis of contrastive unsupervised representation learning. In ICML, 2019.
> >
> > [2] Mahmoud et al. "Masked siamese networks for label-efficient learning." arXiv preprint arXiv:2204.07141 (2022).
> >
> > [3] Li et al. "Masked siamese convNets." arXiv preprint arXiv:2206.07700 (2022).
> >
> > [4] Huang et al. "Contrastive masked autoencoders are stronger vision learners." arXiv preprint arXiv:2207.13532 (2022).
> >
> >
> > **Point b**
> > > Second, the second encoder in the alignment term is one of potentially infinitely many pseudo inverse decoders, and not learned via a contrastive loss, so there is no indication that it will have similar properties to an encoder actually learned using a contrastive objective.
> >
> > As for your second concern on the pseudo-inverse encoder, as discussed above, a small reconstruction loss can also imply better alignment between $f$ and $f_g$. Thus, in MAE, the pseudo-inverse encoder $f_g$ will indeed have aligned features with the real encoder $f$. Though, our main interest in MAE is still the real encoder $f$, and we have provided theoretical guarantees on its downstream generalization (Theorem 4.4).
> >
> >
> > **Point c**
> > > the alignment term on the RHS of the last inequality in (5) (latest revision) vanishes if one assumes the decoder is invertible on its output space (such that $g(f_g(x_2, p_{x_2}), p_{x_2})=x_2$) which is not necessarily an unrealistic property.
> >
> > We are afraid that we are not sure what you mean by saying "the alignment term vanishes". As far as we could understand, when the decoder is invertible, we have $x_2=g(f_g(x_2,p_{x_2}),p_{x_2})$, and in Eq. (5), only the error term $\varepsilon$ vanishes. At this time, we can **directly replace $x_2$ with $g(f_g(x_2,p_{x_2}),p_{x_2})$** and see that the MAE loss is now **equal to the alignment term**, i.e.,
> > $$L_{rec}(f, g) = E_{x_{1}, x_{2}}\Vert g(f(x_{1}, p_{x_{1}}), p_{x_{2}})-x_2\Vert^{2}=E_{x_{1}, x_{2}}\Vert g(f(x_{1}, p_{x_{1}}), p_{x_{2}})-g(f_{g}(x_{2}, p_{x_{2}}), p_{x_{2}})\Vert^{2}.$$
> > Based on this equality, applying the $L$-Lipchitzness of the decoder $g$, we can directly induce that the implicit alignment loss (in the latent space) is an upper bound on the reconstruction loss:
> > $$L_{rec}(f, g)\leq L\cdot E_{x_{1}, x_{2}}\Vert f(x_{1}, p_{x_{1}})-f_{g}(x_{2}, p_{x_{2}})\Vert^{2}.$$
> > Therefore, **our analysis still holds under invertible decoder**.

---

> > ### Author Response · Authors · 2022-08-09
> > **Last chance to discuss**
> >
> > Dear Reviewer qc3H,
> >
> > Thanks again for your response. For your further concerns, we have well-prepared a clarification on the hypothesis. Could you please have a look? Hope we could have the last chance to discuss with you in the last several hours.
> >
> > Have a nice day!

---

### Official Review · Reviewer_8imA · 2022-07-12

**Rating:** 5
**Confidence:** 3
**Soundness:** 3 good
**Presentation:** 2 fair
**Contribution:** 3 good

**Summary:**

The paper presents theoretical results of MAE, a recently popular approach for self-supervised visual representation learning. Two insights are provided: 1) MAE loss can be bounded by an implicit alignment loss, and switching to this loss appears to be helpful for MAE pre-training; 2) masking at a high ratio implicitly bridges different instances of the same class. Experiments are mainly done on small-scale datasets, CIFAR-10 and ImageNet-100.

**Questions:**

- Clarification: how is $f_g$ obtained in the U-MAE loss? According to the theory it seems $f_g$ is a hypothetical function that inverts the decoder, but it is not clear how such a decoder can be trained empirically.
- Rebuttal request: I don't think Figure 3 is convincing enough. It just shows the L2 distances with intra-class examples and intra-instance views are decreasing as the mask ratio decreases. However, it is not clear what's the trend of the *overall* distances in the entire dataset, and what's the trend of the inter-class examples. I think it is more important to show the *relative* trend between intra-class and inter-class, rather than the *absolute* trend.

**Limitations:**

It's a theoretical paper, I do not find the paper addressing limitations and potential negative impact, which is fine to me.

**Strengths And Weaknesses:**

(+) Analyzing and understanding state-of-the-art approaches, especially in self-supervised learning is of great significance.

(+) While I have not checked the math carefully, I find the insights discovered in the paper interesting, and it also appears to be quite well-supported by the experiments.

(-) The paper presents two results, one by treating MAE as implicit contrastive learning (Section 3), and one by building on a theory of generalization. While both are interesting, it is generally making the paper less focused (it could just be two separate works instead).

(-) I don't like the writing style of the paper, where the derivations appear in the main text. I would hope to see clean results, key equations, and detailed insights in the main paper, and leave the derivation part in the appendix.

(-) Figure 1 appears to be not so necessary, as MAE paper has already shown a similar trend on the standard, large-scale dataset ImageNet-1K.

(-) Figure 2 is a bit artificial and less *convincing*. The visualization just shows a particular pair of examples. Many concerns can arise: what if the intra-class variation is so big that no shared, similar views can be found between two examples? what if the shared views are across two examples from *different* classes?

Overall, I think the paper is presenting some interesting results and findings, but I don't think they are solid enough from the perspective of empirical justifications. I do hope my concerns can be addressed so that I am no longer on the fence for this paper.

---

> ### Author Response · Authors · 2022-08-02
> **Response to Reviewer 8imA (2/2)**
>
> **Q6.** In Figure 3, it is not clear what's the trend of the overall distances in the entire dataset, and what's the trend of the inter-class examples. I think it is more important to show the relative trend between intra-class and inter-class, rather than the absolute trend.
>
> **A6.** Following your suggestions, we also add new empirical evidence in the **revised Figure 3**, where we add new results on inter-class distance (b), overall distance (c), and relative distance (intra/inter) (d). We can draw the following observations from Figure 3:
> - Figure 3a, 3b, 3c show that both intra-class, inter-class, and overall distance decrease with larger mask ratio, echoing our analysis in **A4** that a high mask ratio increases sample similarity.
> - Figure 3d shows that relatively speaking, the intra-class distance decreases faster than the inter-class distance under moderate mask ratios, which also aligns well with our hypothesis.
> - In Figure 3d, the relative distance attains its minimum at $\rho\approx0.7$, which is pretty close to the optimal masking rate of MAE is $\rho=0.75$. When $\rho=0.8$, the relative distance becomes larger again, indicating that the difference between intra-class and inter-class edges disappears under too large mask ratio.
>
> Thus, our newly added experiments provide clear evidence of our interpretation of MAE's masking, and the results even agree surprisingly well with the practice of MAE, like the optimal mask ratio.
>
> ---
>
> Hope our explanations and newly added experiments above could address your concerns. We are looking forward to your reply and please let us know if there is more to clarify.

---

> ### Author Response · Authors · 2022-08-02
> **Response to Reviewer 8imA (1/2)**
>
> We thank Reviewer 8imA for appreciating the novelty and solidness of our work. We have fixed the typos following your suggestions. Below, we address your main concerns.
>
> ---
>
> **Q1.** Why are there two theoretical results: implicit contrastive learning (Sec 3) and generalization theory (Sec 4)?
>
> **A1.** The two parts are actually closely connected and complement each other. In particular, Sec 3 relates MAE to implicit contrastive learning, which forms the basis of our analysis, but it alone does not fully explain the effectiveness of MAE. Sec 4 is further built upon the implicit alignment view (Sec 3), and the generalization theory further provides downstream guarantees of MAE that complete the discussion in Sec 3. Thus, the two theoretical results combined contribute to our theoretical understanding of MAE as a whole.
>
>
> ---
>
>
> **Q2.** Writing style needs to be improved.  I would hope to see clean results, key equations, and detailed insights in the main paper, and leave the derivation part in the appendix.
>
> **A2.** Thanks for your suggestions, we reorganize the theoretical part in both Sec 3 and Sec 4. Following your suggestions, we put key theorems and insights in the main paper and defer the derivation to the Appendix. In particular, we have added two theorems (Theorems 3.4 & 4.1) and two assumptions (Assumptions 4.2 & 4.3), accompanied by comprehensive discussions.
>
> ---
>
> **Q3.** Figure 1 appears to be not so necessary, as MAE paper already demonstrates a similar trend.
>
> **A3.** Indeed, as you pointed out, it is not so necessary. So we remove Figure 1(b) in the rebuttal revision following your suggestion.
>
>
>
> ---
> **Q4.** Concerns about Figure 2.
>
> **A4.** We will address your concerns point by point.
>
> **Point a**
> > what if the intra-class variation is so big that no shared, similar views can be found between two examples?
>
> When the intra-class variance is large, we can usually still find intra-class similar views by increasing the mask ratio as long as the intra-class variance is smaller than the inter-class variance. But indeed, the extremely large intra-class variance may decrease the connectivity of the augmentation graph and our theory also describes this situation. In our theory, the extremely large intra-class variance implies the decrease of the connectivity of the augmentation graph, i.e., the decrease of factor $\lambda_{k+1}$ in our proposed guarantee (Theorem 4.4), which means the downstream error increases. So even if the intra-class variance is quite large, our theory can still theoretically analyze the influence of it.
>
> **Point b**
> > what if the shared views are across two examples from different classes?
>
> Indeed, a larger mask ratio also brings a larger risk of inter-class overlapping views while helping bridge intra-class samples together. So there will be a tradeoff between utilizing intra-class and avoiding inter-class connections, and the optimal mask ratio should lie in the middle. We also add some new experiments in **Figure 3** to verify this point, and we defer the discussion to **A6**.
>
> ---
>
> **Q5.** How is $f_g$ obtained in the U-MAE loss?
>
> **A5.** As you pointed out, $f_g$ is a hypothetical function in our discussion. Thus, in the U-MAE loss, we replace it with the encoder $f$, as a large distance between encoder outputs can also effectively promote feature diversity. Then our training objective only involves the encoder $f$ and the decoder $g$, which is easy to jointly optimize. We have made this point explicit in the revision (**Sec 3.2**).

---

> ### Author Response · Authors · 2022-08-06
> **Need further clarification?**
>
> Thanks very much for your constructive and detailed comments. We have tried our best to address the concerns and revised our paper accordingly. Is there any unclear point that we should/could further clarify?

---

> > ### Comment · Reviewer_8imA · 2022-08-08
> > **Thanks for the detailed response, will keep the acceptance rating with one minor suggestion**
> >
> > I would like to thank the hard work on improving the draft (a lot) and the detailed response. Given the new draft and the clarifications, a lot of my concerns and questions are addressed. I do see interesting aspects of the paper (both for hypothesis and for analysis), so I would like to keep my acceptance rating.
> >
> > One minor suggestion: could Fig 3 show intra-class, inter-class and overall distances *in the same figure*? I believe it would be much easier for readers to compare the trend (on top of the "relative" figure shown in (d)).

---

> > > ### Author Response · Authors · 2022-08-08
> > > **Thanks for your reply and we have revised Fig 3**
> > >
> > > Thanks for your kind reply. We are glad to hear that our revisions and responses help address your concerns. We have also reorganized Figure 3 following your advice.
> > >
> > > Sincerely,
> > >
> > > Authors

---

### Official Review · Reviewer_M1jb · 2022-07-13

**Rating:** 4
**Confidence:** 4
**Soundness:** 2 fair
**Presentation:** 2 fair
**Contribution:** 2 fair

**Summary:**

This paper attempts to understand how masked autoencoder (MAE) works and investigates why the masking ratio of MAE is important. On top of the analysis, the paper proposes a uniformity-promoting MAE loss which additionally regularizes the loss by enforcing uniformity. In experiments, the uniformity-promoting MAE shows better performance than the original MAE on CIFAR-10 and ImageNet-100 datasets.

**Questions:**

 - I wonder if the investigation and solution (U-MAE) can be transferred to other masked image modeling methods like BEiT [1], iBOT [2], SimMIM [3], and so on. Discussion about the transferability of those methods and relationships among such masked image modeling methods will be interesting.

[1] Bao et al., Beit: Bert pre-training of image transformers, ICLR 2022
[2] Zhou et al., iBOT: Image BERT Pre-Training with Online Tokenizer, ICLR 2022
[3] Xie et al., SimMIM: A Simple Framework for Masked Image Modeling., CVPR 2022

**Limitations:**

the review will be updated in a couple of days.

**Strengths And Weaknesses:**

- Strengths
	- Theoretical analysis of masked autoencoder gives somewhat new perspective and insight.
	- Proposed uniformity-promoting MAE seems a reasonable loss design and works well on CIFAR-10 and ImageNet-100.
- Weaknesses
	- The experiments are weak, so it is hard to convince the paper's main claim. As in table 1, the experiments are conducted on CIFAR-10 and ImageNet-100 which are not standard for evaluating self-supervised learning methods. It is necessary to conduct at least one comparison with MAE *on ImageNet-1k* as in the original MAE paper.
	- The main formulation and analysis are based upon the MAE framework, so I wonder if the observations and knowledge can be transferred to other masked image modeling methods like BEiT [1], iBOT [2], and SimMIM [3], and so on. Discussion about the transferability of those methods and relationships among such masked image modeling methods will be interesting.
	- It is hard to convince the claim in Figures 2 and 3(a) saying "the higher mask ratio will generate more similar views of intra-class samples." at line 241. Figure 2 seems to cherrypick samples. It is not guaranteed that the surviving patches have similar semantic meanings in natural image datasets like ImageNet. In figure 3, a smaller L2 distance at a higher mask ratio may come from the loss of variety and diversity of image patches.


[1] Bao et al., Beit: Bert pre-training of image transformers, ICLR 2022
[2] Zhou et al., iBOT: Image BERT Pre-Training with Online Tokenizer, ICLR 2022
[3] Xie et al., SimMIM: A Simple Framework for Masked Image Modeling., CVPR 2022

---

> ### Author Response · Authors · 2022-08-02
> **Response to Reviewer M1jb (2/2)**
>
> (continuing A3)
>
> **Point b**
> > In figure 3, a smaller L2 distance at a higher mask ratio may come from the loss of variety and diversity of image patches.
>
> In the revised Figure 3, following your suggestion, we have also taken the feature diversity into consideration by measuring the overall $L_2$ distance between all patches. We can see that the overall distance indeed decreases with a larger mask ratio (**Figure 3(b) and 3(c) (new)**). We also plot the relative distance (intra-class / inter-class) in **Figure 3(d) (new)**, which first decreases with a large mask ratio, attains the minimum at around 0.7, and then increases at 0.9. This shows that relatively, a large mask ratio will first increase intra-class similarity and help feature clustering; when the mask ratio is too large, intra-class similarity decreases because the image semantics are largely distorted. In particular, the sweet spot is $\rho=0.7$, which is close to the optimal mask ratio of MAE, $\rho=0.75$. Thus, this trend also aligns well with the performance of MAE, showing that our augmentation graph analysis indeed helps understand the role of masks in MIM methods.
>
> ---
> **Q4.** Can U-MAE be transferred to other MIM methods, like BeiT, iBOT, and SimMIM?
>
> **A4.** As we discussed in **A2**, our analysis can be easily adapted to other MIM methods with **similar theoretical results**. Due to the limit of time, we take SimMIM as an example, and we follow the official Implementation and the default settings of SimMIM. Similar to MAE, we propose a Uniformity-promoting SimMIM loss (U-SimMIM) which adds a uniformity regularizer term to the original reconstruction loss of SimMIM. We train ViT-Base on ImagNet-100 with 200 epochs. Results are shown below.
>
>
> |Method | Top-1 Linear Accuracy (%)  |
> |---|---|
> |SimMIM|21.59% |
> |**U-SimMIM (ours)**| **30.05% (+8.46%)** |
>
>
> We could see that our U-SimMIM also significantly outperforms SimMIM on large-scale datasets by improving 8.46% Top-1 accuracy on ImageNet-100.
> We have added this result in **Appendix B.6**.
>
> ---
>
> Hope our discussion and newly added experiments could address your concerns. We are looking forward to your reply and please let us know if there is more to clarify.

---

> > ### Comment · Reviewer_M1jb · 2022-08-09
> > **Thanks for the response**
> >
> > Thanks to the authors for their responses!
> > - A2 & A3: Overall I like the responses and discussions.
> > - A1 & A4: It is good to see additional large-scale experiments, but the experimental setting is questionable. The baseline accuracy for MAE and SimMIM looks poor, (e.g., SimMIM's linear prob accuracy of 21.59% is relatively lower than the reported one of 56.7%), so the improvement from such baselines is little convincing. I guess this performance degradation came from a small number of training epochs. I think if the paper has the results with original training settings (800 or 1600 epochs) and they still show a consistent meaningful performance gap, then it would surely be acceptable for the conference.
> >
> > Thanks again to the authors and I will make my final decision after having discussion with other reviewers.

---

> > > ### Author Response · Authors · 2022-08-09
> > > **Thanks for your reply and further response**
> > >
> > > Thanks for your reply and for appreciating our response! We are glad to know that we have addressed your concerns on Q2 and Q3.
> > >
> > > First of all, we want to note that our work is **a theory paper** and our contributions mainly lie in the theory side. As far as we know, we are **the first theoretical attempt to analyze the role of masking in MAE**. Specifically, we establish **a close connection between MAE and contrastive learning** by proving that the MAE loss is upper bounded by an implicit alignment loss, which helps us **understand how masking works in MAE**. Based on this connection, we provide **the first guarantee on MAE's downstream generalization**, and **formally characterize the effect of different masking ratios**. Therefore, we believe that our theoretical analysis of MAE contributes to a better understanding of this new emerging SSL paradigm, and will hopefully inspire more principled research to better design and utilize MIM-style methods.
> > >
> > > The U-MAE experiment is mainly designed as a **proof-of-idea experiment of the theoretical connection between MAE and contrastive learning**. In particular, we aim to show that the uniformity loss of contrastive learning can also be adopted to help improve MAE's linear probing performance. We believe that our experiments on a wide range of datasets, CIFAR-10 (small), ImageNet-100 (medium), and ImageNet-1k (large) indeed **validate the existence and usefulness of this theoretical connection**.
> > >
> > > Below, we further elaborate on our experiments to ease your concerns.
> > >
> > > Following your suggestions, we have tried our best to provide more complete experiment results on ImageNet-100 and ImageNet-1k in the rebuttal phase. However, due to the limit of time and computation resources, we are only able to provide a 200-epoch MAE experiment on ImageNet-1k. We also managed to provide a SimMIM experiment on ImageNet-100. To be consistent with the MAE setup on ImageNet-100 (Sec 5.1), we also train SimMIM for 200 epochs. We will definitely provide more complete results with longer training epochs in the revision with more available time.
> > >
> > > As for your concerns on SimMIM results, because our new experiments on SimMIM are conducted on **ImageNet-100**, we are afraid these results **cannot be directly compared to the reported ImageNet-1k results of SimMIM** as you mentioned. Nevertheless, we indeed **faithfully adopt the official code and follow the default hyperparameters of SimMIM on ImageNet-1k** (https://github.com/microsoft/SimMIM). However, these ImageNet-1k hyperparameters could be less suitable for ImageNet-100, which might help explain the relatively low results on ImageNet-100. Our 800-epoch experiment is still running. For now, we can also draw some clues from the training dynamics of MAE on ImageNet-100 shown in Fig 5b (link https://ibb.co/QjMVBQT). We can see that U-MAE starts to outperform MAE from an early stage, and the gap grows larger and gradually becomes stable. From the trends of two lines, we do not think U-MAE will fail to outperform MAE as the training goes on, as the gap is larger than 10\% which is not easily erased only through longer training.
> > >
> > > ---
> > >
> > > Thanks for your response and hope our explanations above could address your concerns.

---

> ### Author Response · Authors · 2022-08-02
> **Response to Reviewer M1jb (1/2)**
>
>
> We thank Reviewer M1jb for careful reading and appreciating its novelty. We will summarize and address the three main weakness points and the questions you mentioned.
>
> ---
> **Q1.** The experiments are weak. It is necessary to conduct at least one comparison with MAE on ImageNet-1k as in the original MAE paper.
>
> **A1.** Following your suggestion, we add experiments on **ImageNet-1k** and directly adopt the official implementation of MAE. Due to the limit of time, we train **ViT-Base for 200 epochs** with MAE and U-MAE objectives, respectively, and compare their linear probing accuracy to examine feature clustering quality. We adopt the official code of MAE, and for U-MAE, we simply adopt the default hyperparameters without tuning. Results are listed below.
>
> |Method | Top-1 Accuracy (%)  | Top-5 Accuracy (%) |
> |---|---|---|
> |MAE|39.72% | 65.62%
> |**U-MAE (ours)**| **46.49% (+6.77%)** |  **71.92% (+6.3%)** |
>
> We could see that our U-MAE also significantly outperforms MAE on large-scale datasets by improving 6.77% Top-1 and 6.3% Top-5 accuracy on ImageNet-1k.
> We have added this result in **Appendix B.5**.
>
> ---
> **Q2.** Can the existing analysis on MAE be transferred to other MIM methods like BEiT, iBOT, and SimMIM.
>
> **A2.**  The basic paradigm of MIM is to reconstruct the masked patches from unmasked ones, while they mainly differ in implementation details, e.g., 1) **the encoder input**: including masked patches (SimMIM, iBOT, BEiT) or not (MAE), 2) **the decoder**: one-layer (SimMIM, iBOT) or multi-layer (MAE, BEiT); and 3) **the target**: RGB (SimMIM, MAE) or tokenized discrete tokens (iBOT, BEiT). In fact, our theoretical framework is quite general, and can be easily extended to these variants:
> - **Encoder input**. If masked patches are adopted, the input and output could be modeled as $\tilde{x}_1=(x_1,p)$ and $\tilde{x}_2=(x_2,p)$, where they all have access to the entire position encoding.
> - **Decoder**. One-layer case is a special case of our multi-layer formulation. Moreover, in one-layer case, the decoder output of $(\tilde{x}_1,\tilde{x}_2)$ is in an alignment between $f(\tilde{x}_1)$ and decoder weight of $\tilde{x}_2$. Thus, the discussion can be further simplified as we do not require the invertibility of the decoder.
> - **Target.** The tokenized target only corresponds to a specific format of $x_2$, and does not affect our analysis of their implicit alignment.
>
> Thus, our discussion is applicable to different variants of the MIM paradigm.
> Following your advice, we have added a discussion on this point in **Appendix C.1**.
>
> ---
> **Q3.** Concerns about the claim in Figures 2 and 3(a).
>
> **A3.** We address the two concerns point by point.
>
> **Point a**
> > Figure 2 seems to cherrypick samples. It is not guaranteed that the surviving patches have similar semantic meanings in natural image datasets like ImageNet.
>
> Indeed, two **random masks** could generate dissimilar views. However, our point here is to say that a large masking ratio improves **the possibility of finding shared views between samples**. When the raw images are not very similar, like the examples in Fig 2, a small mask ratio can never bridge them together. As the mask ratio grows, there will be more likely to find some "good" masks such that the two samples have similar augmented views, and **through these good masks**, the two intra-class samples can be bridged together. In other words, a large mask ratio brings a large chance of support overlap. This augmentation overlap phenomenon is also discussed and empirically verified in Wang et al [1].
>
> Therefore, when the intra-class variance is large, we can usually still find intra-class similar views by increasing the mask ratio as long as the intra-class variance is smaller than the inter-class variance. But indeed, the extremely large intra-class variance may decrease the connectivity of the augmentation graph and our theory also describes this situation. In our theory, the extremely large intra-class variance implies the decrease of the connectivity of the augmentation graph, i.e., the decrease of factor $\lambda_{k+1}$ in our proposed guarantee (Theorem 4.4), which means the downstream error increases. So even if the intra-class variance is quite large, our theory can still theoretically analyze its influence.
>
>
> [1] Yifei Wang et al., Chaos is a ladder: A new theoretical understanding of contrastive learning via augmentation overlap. In ICLR, 2022.

---

> ### Author Response · Authors · 2022-08-06
> **Need further clarification?**
>
> Thanks very much for your constructive and detailed comments. We have tried our best to address the concerns and revised our paper accordingly. Is there any unclear point that we should/could further clarify?

---

> ### Author Response · Authors · 2022-08-07
> **Comments on the rebuttal?**
>
>
> Dear Reviewer M1jb,
>
> We understand that perhaps you are too busy to read the rebuttal. But since there are only two days left, we are sorry to remind you again. We want to note that we have done the following major updates to address your concerns:
> - In response to your concerns on large scale datasets, we have added new experiments of MAE and U-MAE on ImageNet-1k, where our U-MAE shows significant improvement.
> - We further showed that our methods can be extended to other MIM methods. Our U-SimMIM variant has also obtained significant improvements over SimMIM on ImageNet-1k.
> - Besides, regarding your concerns on Figures 2 & 3, we have elaborated the point of Figure 2, and revised Figure 3 to include more complete results taking patch diversity into consideration. The new results provide further evidence of our theory.
>
> We are wondering if our response satisfies you? We are happy to answer any further questions.
>
>
> Sincerely,
>
> Authors

---

> ### Author Response · Authors · 2022-08-09
> **Your comments are important for us**
>
> Dear Reviewer M1jb,
>
> There are only several hours left for the discussion. We have spent lots of time and effort preparing a very very detailed response. Hope it can address your concerns. Your comments are very important to us. We are greatly appreciated if you could have a look!
>
> Have a nice day!

---

### Author Response · Authors · 2022-08-02
**A Summary of Paper Updates**


We thank all reviewers for the constructive suggestions, which help make this work more complete. Following their suggestions, we
have  made the following major updates to the paper:
- **Sections 3 & 4**: reorganize the theory part by highlighting the main results and moving derivations to Appendix A.1
 - **Section 3.1**: add more clarifications to our notations.
 - **Section 3.2**: move some proofs to Appendix A and add more elaborations on Assumption 3.2.
 - **Section 4.1**: clarify our assumptions (Assumptions 4.2 & 4.3) and add thorough explanations on the insights of our theorems.
 - **Section 4.2**: add **new experiments** on measuring  (1) the inter-class $L_2$ distance, (2) the overall $L_2$ distance and (3) the relative intra-class and inter-class distance. We show that the sweet spot of our analysis aligns well with the practice of MAE.
 - **Section 5.2**: add more explanations on experiment details and more detailed discussions on empirical results.
 - **Appendix B.3, B.4**: add more clarifications on the settings of our experiments.
 - **Appendix B.5, B.6**: complement experiments of U-MAE loss on **ImageNet-1K** and **SimMIM**.
 - **Appendix C.1**: add discussions on the transferability of our theory to other MIM methods.

---

### Meta-Review · Area_Chair_TWs1 · 2022-08-30

**Recommendation:** Accept
**Confidence:** Less certain

**Metareview:**

The authors aim to shed some light on the recent success of masked autoencoders (MAEs) by theoretically analyzing the MAE loss and the role of high masking ratios. In particular, the authors show that, under some assumptions, the MAE loss can be upper-bounded by an implicit alignment loss. The authors argue that this connection to contrastive self-supervised learning is critical and allows one to formally characterize the effect of different masking ratios and establishing a generalization bound. Then, the authors propose a regularizer based on the analysis and empirically evaluate its effectiveness.

The reviewers appreciated the timeliness and significance of this work. The insights brought forward seem relevant and important enough to the larger community focused on the theoretical underpinnings of MAEs. Two main weaknesses were identified: (1) Does this potentially very loose upper-bound actually allow one to make the conclusions in this work? After all, the models in practice might not satisfy the stated assumptions. In fact, the transformer architecture itself might be the driving force behind the success. (2) The empirical validation is not ideal -- given that the authors didn't reproduce the original results on ImageNet-1k it's unclear whether the proposed regularizer improves the results on top of the non-regularized MAE.

After the discussion phase the ratings remained borderline and two reviewers were still not fully convinced. I personally feel that both (1) and (2) were partially addressed during the discussion. Having in mind that providing definite results in this setting is incredibly challenging, I find that this work is a valuable contribution to the community of researchers trying to improve the understanding of the underlying mechanisms of self-supervised learning.

**Award:**

No

---

### Decision · Program_Chairs · 2022-09-14

Accept